# ConvoSource: Radio-Astronomical Source-Finding with Convolutional Neural Networks

**Vesna Lukic \*, Francesco de Gasperin and Marcus Brüggen**

Hamburg Observatory, University of Hamburg, Gojenbergsweg 112, 21029 Hamburg, Germany; fdg@hs.uni-hamburg.de (F.d.G.); mbrueggen@hs.uni-hamburg.de (M.B.)

**\*** Correspondence: vesna.lukic@hs.uni-hamburg.de

**Abstract:** Finding and classifying astronomical sources is key in the scientific exploitation of radio surveys. Source-finding usually involves identifying the parts of an image belonging to an astronomical source, against some estimated background. This can be problematic in the radio regime, owing to the presence of correlated noise, which can interfere with the signal from the source. In the current work, we present ConvoSource, a novel method based on a deep learning technique, to identify the positions of radio sources, and compare the results to a Gaussian-fitting method. Since the deep learning approach allows the generation of more training images, it should perform well in the source-finding task. We test the source-finding methods on artificial data created for the data challenge of the Square Kilometer Array (SKA). We investigate sources that are divided into three classes: star forming galaxies (SFGs) and two classes of active galactic nuclei (AGN). The artificial data are given at two different frequencies (560 MHz and 1400 MHz), three total integration times (8 h, 100 h, 1000 h), and three signal-to-noise ratios (SNRs) of 1, 2, and 5. At lower SNRs, ConvoSource tends to outperform a Gaussian-fitting approach in the recovery of SFGs and all sources, although at the lowest SNR of one, the better performance is likely due to chance matches. The Gaussian-fitting method performs better in the recovery of the AGN-type sources at lower SNRs. At a higher SNR, ConvoSource performs better on average in the recovery of AGN sources, whereas the Gaussian-fitting method performs better in the recovery of SFGs and all sources. ConvoSource usually performs better at shorter total integration times and detects more true positives and misses fewer sources compared to the Gaussian-fitting method; however, it detects more false positives.

**Keywords:** deep learning; radio astronomy; source-finding; methods; analysis

---

## 1. Introduction

An ongoing task in astronomy is the ability to find astronomical sources. This is of importance because it forms the basis by which a radio astronomical catalog can be built. Modern radio telescopes can observe many millions of radio sources, and this number will only increase in time owing to rapidly developing technologies [1]. It is therefore important that the methods developed to find sources can keep up with the capabilities of the technology, with respect to the quality of sources that are detected by the telescope.

In this section, we give a brief summary of the main factors affecting the ability to find sources in radio data, the different types of radio sources (star forming galaxy (SFG) or type of AGN), how a machine learning approach can work, details about the simulated Square Kilometer Array (SKA; [2]) data used, as well as a brief review of the previous work in this area.

We note that the paper focuses only on source-finding, where the output of the algorithm is the predicted location of a source. Characterizing and classifying the sources is not currently explored.

### 1.1. Source-Finding at Radio Frequencies

Radio telescopes measure the surface brightness of the radio sky across some frequency or range of frequencies and produce a map of the surface brightness. What constitutes a source is a collection of pixels above some value, which is determined by estimating the background, or noise. The noise is usually composed of a combination of instrumental noise, observed background emission, and leftover system uncertainties [3].

The first step involved in source-finding is usually pre-processing the image containing the radio sources. This involves some transformations to the image, such as scaling the pixel intensities, to facilitate the source-finding method by suppressing undesired distortions or enhancing features [4], while preserving the physics of radio sources in the image. The second step is to estimate the background, after which a threshold can be chosen, which defines where the sources are. Contiguous pixels above a certain threshold are considered to form part of an object [5], after which a local peak search is performed where maximum value pixels are isolated.

In the presence of low SNR, which occurs when there is a relatively high background compared to surface brightness (signal) from the source, it can be difficult to group the pixels belonging to a particular source. Additionally, the sizes and intensities of the astronomical bodies can vary significantly [6]. As the SNR is increased, finding and extracting the sources becomes easier as the pixels belonging to the source show a greater contrast compared to the background. However, it is more frequently the case that shorter integration times are used, which results in noisier data, and it is not always easy to capture the background signal, which may also vary across regions in the image. Another problem to consider is that of source confusion, which is the inability to measure faint sources due to the presence of other sources nearby. Furthermore, at radio frequencies, the noise tends to be more correlated compared to other frequencies [7–9], posing further challenges for source-finding and extraction.

Many algorithms have been developed to perform source-finding across different wavelengths such as optical, radio, infrared, or x-ray, some of which use a combination of techniques. Masias et al. [10] presented the largest overview of the most common techniques, although there have been more recent developments. For example, a source extractor originally developed for source-finding in optical images (ProFound; Robotham et al. [11]) can also successfully be used at radio wavelengths [9].

One state-of-the-art source-finding algorithm is the Python Blob Detector and Source-Finder[1] (PyBDSF; [12]), which works as follows: After reading the image, it performs some pre-processing, for example computing the image statistics. Using a constant threshold for separating the source and noise pixels, the local background rms and mean images are computed. Adjacent islands of source emission are identified, after which each island is fit with multiple Gaussians or Cartesian shapelets. The fitted Gaussians or shapelets are flagged to indicate whether they are acceptable or not. The residual FITSimages are computed for both Gaussians and shapelets. Gaussians within a given island are then grouped into discrete sources.

There is a growing number of works on the use of machine learning, particularly deep learning methods, in the detection of astronomical objects. For example, González et al. [13] used a deep learning framework and a real-time object detection system to detect and classify galaxies automatically, including a novel augmentation procedure. Ackermann et al. [14] investigated the use of deep convolutional neural networks (CNNs) and transfer learning in the automatic visual detection of galaxy mergers and found them to perform significantly better than current state-of-the-art merger detection methods. An outlier detection technique has also been developed using an unsupervised random forest algorithm and found to be successful in being able to detect unusual objects [15]. Gheller et al. [16] developed COSMODEEP, a CNN to detect extended extragalactic radio sources in

---

[1]    https://www.astron.nl/citt/pybdsf/.

existing and upcoming surveys, which proved to be accurate and fast in detecting very faint sources in the simulated radio images. There have been a couple of recent works specifically in the area of finding radio sources. ClaRAN (Classifying Radio sources Automatically with Neural networks) [17] trained a source-finder on radio galaxy zoo data [18] to learn two separate tasks, localization and recognition, after which the source was classified according to the number of peaks and components, with accuracies >90%. Sadr et al. [19] presented DeepSource, a CNN that additionally used dynamic blob detection to find point sources in simulated images and compared the results against PyBDSF, using different signal-to-noise ratios. In contrast, the current work examines the recovery of SFGs and two classes of AGN, as well as all sources combined, at different SNRs using a CNN architecture we call ConvoSource, compares the results against PyBDSF, and shows in which circumstances one performs better than the other and the likely reasons why. DeepSource requires the tuning of more variables that need to be defined prior to applying the algorithm, also known as hyperparameters. ConvoSource requires only the usual deep learning parameters such as the number and type of layers and the batch size, in addition to the usual components of a machine learning model, such as a cost function to measure the models' performance and the gradient descent method, which minimizes the cost function.

### 1.2. Types of Radio Sources

Galaxies in the Universe that exhibit significant radio emission generally fall into one of two main categories: SFGs or AGN. Radio-loud AGN can be grouped based on their appearance; they can be either "compact" or "extended". The two most influential factors that govern whether a source will appear point-like, elongated, or very resolved are the distance of the source and the resolution. Different radio source types can be characterized by a different spectral index $\alpha$, which is related to the frequency $\nu$ and flux density $S$ through $S(\nu) \propto \nu^{\alpha}$. The slope of the spectrum is determined by the electron energy distribution. Extended radio sources generally have a steep radio spectrum (typical values are $\alpha \lesssim -0.8$ [20]) and can be referred to as steep-spectrum AGN (AGN-SS), where the majority of sources can be divided into two distinct classes depending on the morphology of the radio lobes: FRI (Fanaroff-Riley type 1; core-dominated) and FRII (Fanaroff-Riley type 2; lobe-dominated) [21]. Compact radio sources tend to exhibit a flat radio spectrum (typical values are $\alpha \leq -0.5$) and are denoted as flat-spectrum AGN (AGN-FS) [22]. It should also be noted that some steep-spectrum sources can be compact.

Since the relative strength of the emission from radio sources depends on frequency, different components of a radio source can have different spectral shapes.

### 1.3. Deep Learning

Deep learning methods have been successful in extracting information from high-dimensional data such as images [23–25]. CNNs are a common example of such a deep learning method. A more detailed description of how CNNs work is given in Section 2.

The current work explores a novel approach to source-finding by training a CNN on a solution map derived from knowledge of the source locations. For our purpose of source-finding, the output images we aim to produce are those of the locations of the sources, rather than the original input source maps. Given that the source locations can be transformed into image data, the source location map, along with the original source map, can be segmented into smaller square images (having a size of 50 × 50 pixels in the current work), which are then used as the inputs to train the CNN to predict the source locations.

### 1.4. Simulated SKA Data

The SKA aims to be the largest radio telescope built to date. It will eventually have a collecting area of more than one square kilometer, operate over a wide range of frequencies (50 MHz–14 GHz in the first two phases of construction), and will be 50 times more sensitive than any other radio

instrument to date. In the meantime, it is possible to use simulated data products to generate data similar to what would be expected to be observed by the SKA. The SKA Data Challenge 1[2] (SKA SDC1; [26]) was a recent challenge set for the community to develop or use existing source-finders to perform source-finding, characterization of the sources, and source population identification (SFG, SS, or FS).

Catalogs of objects to be included in the simulated maps were generated using the Tiered Radio Extragalactic Continuum Simulation (T-RECS) simulation code [27]. The radio sky was modeled in continuum, over the 150 MHz–20 GHz range, with two main populations of radio galaxies: AGN and SFGs and their corresponding sub-populations. The wide ranging frequency has been enabled by allowing specific conditions for the spectral modeling. Across the AGN, the sources were allowed to have a different spectral index below and above ~5 GHz, constrained by the modeled counts from Massardi et al. [28] for the lower frequency range and de Zotti et al. [29] for the higher frequency range. In the SFG population, the spectral modeling included synchrotron, free-free, and thermal dust emission, all expressed as a function of the star forming rate. The redshift range of the simulation was $z = 0$–8. The T-RECS simulation output used for SDC1 contained all the sources in a $3 \times 3$ field of view (FoV) with integrated flux density at 1.4 GHz $> 100$ nJy [27].

The data used in the current work were based on the simulated data products generated for SDC1. There were three available frequencies (B1: 560 MHz, B2: 1400 MHz, and B5: 9200 MHz) at three integration times (8 h, 100 h, and 1000 h) for each frequency. There were nine maps altogether in the form of FITS files. The size of the maps was $32,768 \times 32,768$ pixels; however, only a $4000 \times 4000$ area for B1 and a $4200 \times 4200$ area for B2 within these maps (the training set area) contained the true source locations. The FoV was chosen for each frequency to contain the primary beam for a single telescope pointing out to the first null, giving a map size of 5.5, 2.2, and 0.33 degrees on a side for B1, B2, and B5, respectively, with corresponding pixel sizes of 0.60, 0.24, and 0.037 arcsec. The properties of sources in a training set area were also provided, across the three frequencies, to see how the sources were characterized in a particular area so the source-finders could be calibrated or trained. We used the generated data and focused on the source-finding aspect only.

In constructing the SDC1 image corresponding to the T-RECS source catalog, sources were injected with a different procedure depending on whether they were extended or compact (major axis greater or smaller than three pixels, respectively) with respect to the adopted frequency dependent pixel size. The SFGs were modeled using an exponential Sersic profile [30], projected into an ellipsoid using a given axis ratio and position angle. The AGN populations (SS and FS) were treated as the same object type viewed from a different angle. SS AGN would assume FRI/FRII morphologies, and FS AGN were composed of a compact core with a single lobe, but pointing in the direction of view. The steep-spectrum sources were generated as postage stamps (that included affine transformations) from a library of scaled real high resolution images. They also had a correction applied to the flux of the core in order to give it a flat spectral index; thus, the same AGN could have a different core to lobe fraction when viewed at different frequencies. The FS AGN were added as a pair of circular Gaussian components: a compact core with a more extended end-on lobe.

A mild Gaussian convolution was applied to the extended source images, using an FWHM of two pixels. The three catalogs (SFGs, SS, and FS AGN) of compact objects were added to the image as elliptical Gaussian components.

All the compact sources that belonged to the classes of SFGs, SS, and FS AGN were described by an integrated flux density and a major and minor axis size. The compact FS AGN were additionally described with a core fraction that indicated the proportion of emission belonging to the core of the source compared to the source extent.

---

2　https://astronomers.skatelescope.org/ska-science-data-challenge-1/.

Visibility data files were generated using the SKA1-Mid configuration. There were two cases explored: (1) when the 64 Meerkat dishes were included, there were 197 antenna locations specified at B2; and (2) when the Meerkat dishes were not included, 133 antenna locations were used at B1 and B5. Both cases were frequency dependent and reflected the fact that Meerkat would most likely not be equipped with feeds for B1 and B5.

The visibility sampling was based on 91 spectral channels that spanned a 30% fractional bandwidth, using a time sampling that spanned −4 h to 4 h of local sidereal time with an increment of 30 s integration time at an assumed declination of −30. The visibility files were used to generate the noise images and the point spread functions. The gridding weights for the visibility data were determined by firstly accumulating the visibility samples in the visibility grid with their natural weights. After this, an FFT based convolution was applied to the visibility density grid using a Gaussian convolving function with an FWHM of 178 m. The convolving function width was manually tuned to match as closely as possible the sampling provided by the array configuration. Uniform weights for the visibilities were formed by using the inverse of the local smoothed data density. After this, a Gaussian taper was used such that it resulted in the most Gaussian possible dirty beam with a target FWHM of (1.5, 0.60, and 0.0913) arcsec at (560, 1400, and 9200) MHz. The actual dirty beam dimensions were closely matched to the target specification. There was a degradation of image noise compared to the naturally weighted image noise; therefore, they were rescaled in amplitude to represent realistic variations in RMS for the different integration times. Adding the various noise images to the convolved sky model resulted in the final data products.

Additional files provided include the primary beam images, which were used to correct the flux values in the original maps, the synthesized beam images, and the training set files, which included the properties of the sources such as flux, size, and class, for a particular area in the entire map. There were three training set files, for the three frequencies. Therefore, the same training set file was used across the three different integration times within one frequency. For more specific details on the generation of the simulated SKA data, please refer to Bonaldi and Braun [26].

The paper is outlined as follows: In Section 2, we discuss the specifics about the SKA simulated dataset, the pre-processing steps on the raw data, the parameters by which PyBDSF was run, how the dataset was generated prior to undertaking source-finding with ConvoSource and PyBDSF, the theory behind CNNs, as well as how the images were augmented for ConvoSource. Section 3 describes the major results summarized in F1 scores that combine precision and recall. We also provide confusion matrices for some data subsets. Section 4 summarizes our overall findings. Appendix A contains the precision and recall classification metrics.

## 2. Methods

### 2.1. Convolutional Neural Networks

The most common example of a deep learning framework is a CNN. They use convolutional layers, which apply a convolution operation to the input and pass the output to the following layer, ultimately achieving a hierarchical extraction of features. The convolutional layers use filters, whose purpose is to scan across the images and detect features. The filters typically have sizes of a few pixels across and greatly reduce the number of parameters compared to the fully connected layers in traditional neural networks. The reduction of parameters in CNNs helps to avoid the vanishing gradient problem, where the addition of layers can cause the gradient to decrease to zero, a problem most commonly encountered in fully connected neural networks. The filters also ensure parameter sharing, which enforces translational invariance [31], defined as when the network produces the same response if the input is translated horizontally or vertically.

The main purpose of ConvoSource was to find sources in radio astronomy data. The key idea was to train a CNN using input maps and the corresponding solution maps, to reconstruct the solution maps. In the testing stage, only the real maps were needed as an input to the CNN, and the source

locations were predicted using the weights that minimized the cost function. The predicted source locations were compared to the true source locations to calculate the precision and recall metrics. The sources of varying sizes and emission patterns were collapsed into individual pixel locations, and the remainder of the image was blank. ConvoSource was trained to do source-finding using segmented real maps and the corresponding solution maps, both having sizes of $50 \times 50$ pixels, across three SNRs of 1, 2, and 5, and the results were compared with the sources found using PyBDSF.

It should be noted that this method of source-finding can be posed as an image-to-image translation problem, as are many problems in the computer vision field [32]. Many such problems result in an output image having equivalent or greater complexity compared to the input image, requiring the use of a bottleneck (a compressed representation of the inputs) in the architecture. Our method did not require a bottleneck since it produced images of much lower complexity compared to the input maps, where the radio emission of sources was collapsed to between one and a few pixels. We were not attempting to see whether the original map could be reconstructed from the input data using a lower dimensional projection, but to train the network to predict the location of the sources. The stacked convolutional layers extracted the signal from the noise, where each layer produced an output having the same dimensions as the input, in order to directly see the detected signals that were propagated through the network. As a result, we required only the use of a simple CNN.

Another possible application of CNNs on radio astronomy data of the type explored in the current work is to reconstruct the original input maps that contain the sources, as well as the background noise, which is an undesired feature. A better application could be to investigate whether it is possible to derive maps similar to the 1000 h maps using the 8 h emission maps, because the shorter integration time maps can be viewed as noisier versions of the longer integration time maps. This can be the subject of a future work.

The present work used Keras[3] with the TensorFlow[4] backend and Python Version 2.7.15. We used a convolutional network architecture of three consecutive convolutional layers and one dense layer, having a total of 32,193 parameters.

Early stopping was used with a patience of 5 training epochs. A single training epoch was when all training samples were passed through the network. Eighty percent of the data was used for training, and the remaining 20% was used for testing.

The ConvoSource architecture, as shown in Table 1 and Figure 1, was made up of 3 convolutional layers and one dense (fully connected) layer. There were 16, 32, and 64 filters, with a filter size of 7, 5, and 3 in the first, second, and third convolutional layers, respectively. A dropout layer using a dropout fraction of 0.25 was inserted between the first and second convolutional layers to make the network more robust by reducing overfitting. The purpose of the fully connected layer was to integrate the features extracted from the feature maps in the final convolutional layer, in order to output a prediction for the location of the source. We slid the filters along by one pixel in each layer to ensure maximal information extraction. The batch size was set to 128. We used the Adadelta optimizer [33] with a default learning rate of 1.0, decay of 0, and a rho of 0.99. Adadelta is based on Adagrad [34] (an optimizer with parameter specific learning rates); however, Adadelta adapts the learning rates based on a moving window of gradient updates. We also used the binary cross-entropy cost function [35] shown in Equation (1):

$$-\frac{1}{N} \sum_{i=1}^{N} y_i log(\hat{y}_i) + (1 - y_i) log(i - \hat{y}_i), \tag{1}$$

where $y_i$ represents an individual pixel in the solution map, $\hat{y}_i$ represents an individual pixel in the predicted map, and $N$ is the number of pixels in an image. The binary cross-entropy cost function was

---

3 https://keras.io/preprocessing/image/.
4 https://keras.io/losses/#categoricalcrossentropy.

used on the individual pixel values in each solution and corresponding prediction map and summing on a per-image basis, then adding these values across all images in a particular batch. The architecture shown in Figure 1 also contains an example of a real input map and a solution map, the features detected, and the corresponding reconstructed map.

We experimented with using pooling layers, by applying these to the B1 8 h dataset with no augmentation. Pooling reduces the dimensionality of the layer by outputting the average pixel value across some area whose size is defined by the user. Two different architectures were considered: placing a pooling layer after the first or after the second convolutional layer, respectively. Due to the halved dimensions in the architecture as a result of pooling, an upsampling layer had to be inserted prior to obtaining the output. In both cases, the resulting metrics were all inferior to the equivalent model without pooling. The source locations tended to be less precise and generally spanned an area of four square pixels, most likely because the pooling operation lost the precise source location.

The use of pooling resulted in ConvoSource identifying no true positives, and it generated a few false positives due to the reconstruction of the source positions along the edges of the image only. The likely explanation was that the true signal from the source only occupied a small area, therefore, when pooling is used, it could "wash out" these pixels, in some cases causing the source to become lost among the background.

**Table 1.** Architecture of the ConvoSource model.

| Layer | Output Shape | # Params |
|---|---|---|
| Input_1 | (None, 50, 50, 1) | 0 |
| conv2d_1 | (None, 50, 50, 16) | 800 |
| dropout_1 | (None, 50, 50, 16) | 0 |
| conv2d_2 | (None, 50, 50, 32) | 12,832 |
| conv2d_3 | (None, 50, 50, 64) | 18,496 |
| dense_1 | (None, 50, 50, 1) | 65 |
| Total | | 32,193 |

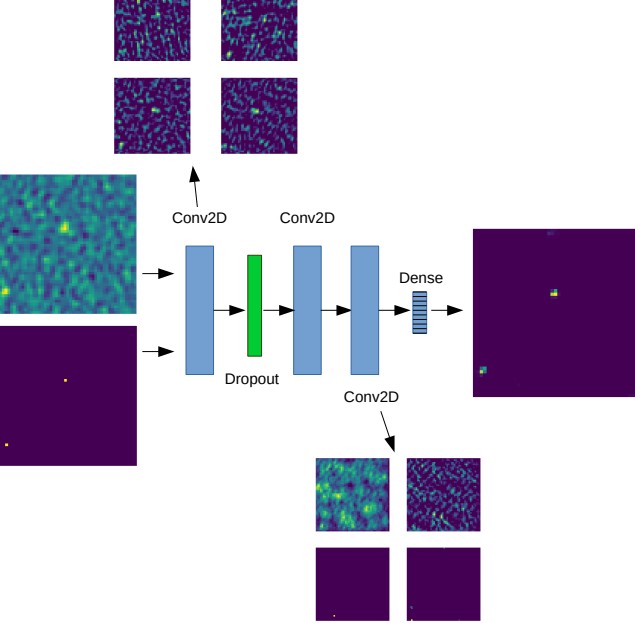

**Figure 1.** ConvoSource architecture and examples of inputs when training (real maps and solution maps), features detected at the output of the first and third convolutional layer, as well as the resulting reconstructed image of the solution map. During testing, only the real maps are input into the network, and the predictions are given using the weights from the trained network.

## 2.2. Pre-Processing

In order to ensure accurate flux values, we used the primary beam image and raw FITS files provided and ran CASA [36] to regrid the image and correct for the primary beam. The resulting FITS file was the one used to perform source-finding in ConvoSource and PyBDSF. Some tests were done using the non primary beam corrected FITS files, and we observed no change in performance regarding source-finding ability. To determine the background noise level in the image, we output the background rms maps when we ran PyBDSF, by specifying RMS_MAP = TRUE using the PROCESS_IMAGE command. To perform source-finding in PyBDSF, we ran the PROCESS_IMAGE command using the default parameters of THRESH_ISL = 3.0 and THRESH_PIX = 5.0.

## 2.3. Dataset Generation

The true location (solution) maps were generated using the training set files across each frequency. Since we were only interested in the source-finding, we took note of the corresponding (x,y) positions of each source in the training set. We focused only on the sources that could be found given the noise. The source locations were inserted into the solution maps as single pixels, under the condition that the sources had a flux above a certain threshold (when the sources had a flux greater than one, two, and five times the mean noise level, referred to as SNR = 1, 2, and 5.)

The solution maps used for training had all the sources encoded with a 1, irrespective of class. When testing, the SS, FS, and SFG sources were encoded using the integers 1, 2, and 3, respectively, in the solution map, in order to calculate how well ConvoSource and PyBDSF recovered each of these classes of sources. Table 2 shows the number of each class of sources across SNR = 2 and SNR = 5, respectively. It should be noted that there were many more SFG sources compared to SS and FS sources, which was why we focused on augmenting those source types to see if this improved the performance of ConvoSource. There were fewer sources available at higher SNRs compared to at lower SNRs, since the threshold for inserting sources into the solution map was a lot higher.

Given there were very few sources available in the B5 dataset, as shown in Table 2, we focused our attention on the B1 and B2 datasets only.

**Table 2.** The total number of steep-spectrum (SS) AGN, flat-spectrum (FS) AGN, and SFGs across each integration time across all frequencies, when using SNR = 2 and SNR = 5.

| Dataset | # SS-AGN | # FS-AGN | # SFG |
|---|---|---|---|
| SNR = 2 | | | |
| B1 | | | |
| 8 h | 342 | 117 | 13,920 |
| 100 h | 644 | 386 | 34,158 |
| 1000 h | 957 | 682 | 57,797 |
| B2 | | | |
| 8 h | 91 | 64 | 4028 |
| 100 h | 166 | 151 | 9423 |
| 1000 h | 278 | 294 | 17,283 |
| B5 | | | |
| 8 h | 3 | 1 | 26 |
| 100 h | 4 | 2 | 103 |
| 1000 h | 6 | 6 | 223 |
| SNR = 5 | | | |
| B1 | | | |
| 8 h | 213 | 94 | 5717 |
| 100 h | 395 | 208 | 16,885 |
| 1000 h | 605 | 366 | 31,597 |
| B2 | | | |
| 8 h | 59 | 25 | 1877 |
| 100 h | 101 | 73 | 5096 |
| 1000 h | 178 | 155 | 10,251 |
| B5 | | | |
| 8 h | 3 | 1 | 7 |
| 100 h | 4 | 1 | 43 |
| 1000 h | 4 | 3 | 114 |

We verified that the noise level in the maps was uniform. Table 3 shows that there were only small proportional differences in the number of solutions obtained when taking the individual quartile cut-offs versus using the cut-offs derived from the whole training set area.

**Table 3.** Percentage difference in the number of sources depending on whether the quartile threshold from the training set was taken versus using the threshold obtained from the training set as a whole, at an SNR = 5.

| Frequency | 8 h | 100 h | 1000 h |
|---|---|---|---|
| B1 | 4.4 | 3.7 | 2.6 |
| B2 | 4.5 | 3.6 | 2.8 |

The left panels of Figures 2 and 3 show a section from a real map of B1 at 1000 h and B2 at 8 h, respectively, containing SFGs, SS, and FS sources, along with the solutions injected at an SNR of 2 and 5. The smaller the SNR, the more sources would appear in the solution map, which would look increasingly less obvious as they would be getting mixed with the noise background. Conversely, the larger the SNR, the fewer sources in the solution map, and only increasingly large and/or bright sources would appear.

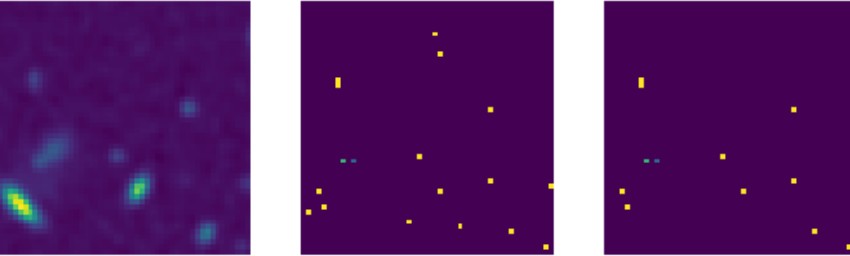

**Figure 2.** (Left panel) Real map of a panel containing a combination of SFGs, SS, and FS sources at B1 at 1000 h. (Middle panel) True source locations at SNR = 2. (Right panel) True source locations at SNR = 5. The yellow, blue, and green pixels indicate SFGs, SS, and FS sources, respectively. In this particular case, both the SS and FS sources are very close together and very faint, which presents a challenge for both source-finders. The panels have a side length of 50 × 50 pixels.

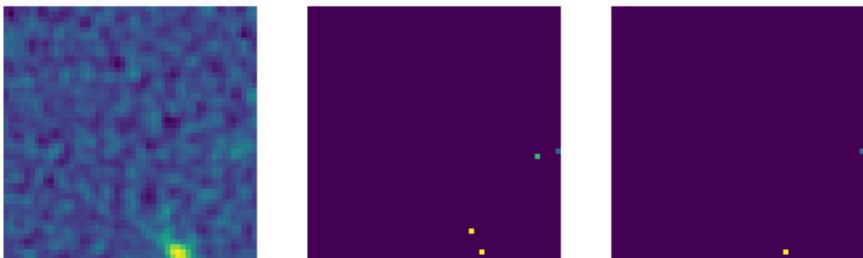

**Figure 3.** (Left panel) Real map of a panel containing a combination of SFGs, SS, and FS sources at B2 at 8 h. (Middle panel) True source locations at SNR = 2. There are two SFGs and one each of SS and FS galaxies. (Right panel) True source locations at SNR = 5. At this SNR, only one SFG and one SS source remain. The other SFG and FS sources had a total flux that was lower than the cut-off threshold at that SNR. The yellow, blue, and green pixels indicate SFGs, SS, and FS sources, respectively. The panels have a side-length of 50 × 50 pixels.

To generate input image data for ConvoSource, we divided the area containing the source locations (the training set area of size 4000 × 4000 pixels for B1 and 4200 × 4200 pixels for B2) into 50 × 50 pixel blocks and moved these blocks across by increments of 50 pixels (resulting in 6400 images), ensuring that the segmented blocks covered the entire area. The blocks may contain sources located on their boundaries; however, all parts of the sources were accounted for. For a discussion of how we decided on the choice of block and increment sizes, please refer to Section 2.3.1 at the end of this section.

In the 4000 × 4000 pixel area at the B1 frequency, there were 6400 images for training and testing altogether, with 5120 (80%) for training and 1280 (20%) for testing. In the 4200 × 4200 pixel area at the B2 frequency, there were 7056 images for training and testing altogether, with 5644 (80%) for training and 1412 (20%) for testing. We also investigated how much the results could be improved when using image augmentation, so 5120 and 5644 were the minimum number of images with which we trained, for B1 and B2, respectively. Similarly, we generated the input solution data for ConvoSource by inserting individual pixels to represent the true location of the source, with the position obtained from the training set.

Given that the pixel values were the surface brightness, which could be very small with $O(10^{-6})$ in magnitude, the image data were multiplied by $10^6$ in the same units. We note that the pixel values could also be negative. Applying a linear scaling to the original values ensured there was sufficient contrast between them, which facilitated detection by the CNN. The scaling was also done to match the order of magnitude of the values in the solution maps, which were generated by inserting a "1" against a background of "0". We note that we also experimented with multiplying the data by $10^9$ and found no noticeable difference.

We used only the training set region out of the whole map, which consisted of a 4000 × 4000 pixel area across the B1 and B5 datasets and 4200 pixel area across B2, as shown in Table 4. It should be

noted that the same area was not covered between the three frequencies; however, it was the same within the same frequency between the three integration times.

**Table 4.** The x and y ranges of the training area, according to the locations within the whole map.

| Frequency | x Range | y Range | Area |
|---|---|---|---|
| B1 | 16,300–20,300 | 16,300–20,300 | 4000 pixels sq. |
| B2 | 16,300–20,500 | 16,300–20,500 | 4200 pixels sq. |

The solution maps were generated in the same way as the input image maps, using $50 \times 50$ pixel blocks with increments of 50 pixels, where a "1" was inserted at the location of the centroid position of the source. The blocks that contained no solutions were empty $50 \times 50$ blocks. The source selection was subject to a flux threshold, where only sources having a flux greater than 1, 2, or 5 times the background for each map were selected. The background maps were determined using PyBDSF. Figure 4 shows the segmentation of part of the training area into $50 \times 50$ pixel blocks, for both the original primary beam corrected FITS file, as well as the corresponding solution map generated. We ran PyBDSF with the default parameters in order to perform source finding across the whole map, which was later made into a subset to only include the training set area in the images.

2.3.1. Effect of Differences in Segmenting the Dataset

There was a trade-off with grid size, the number of images produced, training time, and capturing the radio emission from sources within the blocks. There were two main cases to consider: whether or not there was an overlap between the blocks, which was determined by the increment size relative to the block size. Table 5 details the choice of block and increment size on the number of images produced.

Using more training images and/or using larger block sizes increased the training time. First, we considered the case that there was no overlap between the blocks. We ideally wanted a grid size that would fit into the $4000 \times 4000$ area for B1 and the $4200 \times 4200$ area for B2, without excluding any pixels. Using $20 \times 20$ pixel blocks appeared to be too small: it sometimes caused the sources near the boundary in a block to leak the emission into the adjacent block(s). Additionally, the training time would increase due to having 40,000 images, although the smaller block size would help to decrease the training time. On the other hand, using $200 \times 200$ pixel blocks was a good size in regard to containing the radio emission in a block; however, it would also be more computationally intensive per block and would produce only 400 training images. In the current work, we found the $50 \times 50$ pixel blocks using 50 pixel increments to be optimal in terms of block size, number of images produced, and training time.

The second case considered was when the increments were less than the side-length of a block, causing overlap between the blocks. This meant that parts of the emission in one block would be seen in at least another block. We experimented with using 20 pixel increments (resulting in 39,204 images) instead of 50 pixel increments, such that the same part of a source was seen across at least one other block, and therefore, sources on the boundary in one block would not be on the boundary in a neighboring one, noting a much longer training time with no significant improvement in results.

We note that although the training time would be longer in some of the cases mentioned above, it would only be done the one time, for each SNR, frequency, and exposure time.

 

**Figure 4.** (**Left**) Segmentation of a portion of the primary beam corrected images in the training set area. (**Right**) Segmentation of the solution map in the same area. These images are generated from the B1 1000 h dataset, using an SNR = 5 to determine the threshold of flux for injecting the solutions. Each block formed a single $50 \times 50$ pixel image that was input into the ConvoSource algorithm. The blocks on the left make up the training set images (train_X), and the blocks on the right make up the solution set images (train_Y).

**Table 5.** Exploring different block sizes, pixel increments, and number of images produced, using the B1 frequency as an example.

| Block Size | Increment Size | # Images Produced |
|---|---|---|
| No overlap | | |
| $20 \times 20$ | 20 | $200 \times 200 = 40{,}000$ |
| $50 \times 50$ | 50 | $80 \times 80 = 6400$ |
| $80 \times 80$ | 80 | $50 \times 50 = 2500$ |
| $100 \times 100$ | 100 | $40 \times 40 = 1600$ |
| $200 \times 200$ | 200 | $20 \times 20 = 400$ |
| Overlap | | |
| $20 \times 20$ | 10 | $398 \times 398 = 158{,}404$ |
| $50 \times 50$ | 20 | $198 \times 198 = 39{,}204$ |
| $50 \times 50$ | 40 | $99 \times 99 = 9801$ |
| $80 \times 80$ | 40 | $98 \times 98 = 9604$ |
| $80 \times 80$ | 50 | $79 \times 79 = 6241$ |

*2.4. Image Augmentation*

Deep learning techniques are able to take advantage of image augmentation as it generates more training samples, which should improve the performance up to some threshold [23]. Since there were many fewer SS and FS AGN sources compared to SFGs, we wanted to see whether we could improve on the metrics for these types of sources if we augmented the images that contained them. We employed vertical and horizontal flipping and rotation by 90, 180, and 270 degrees. The results showed the metrics when applying no augmentation, augmenting the SS and FS sources, as well as augmenting all sources. There would be little merit in explicitly augmenting the SFGs because they tended to appear more point like.

**3. Results**

The results presented were the summary metrics of the sources across the different classes, SS, FS, and SFG sources and all sources as a whole. We allowed a leniency of three pixels for the positions of sources found. To calculate the metrics, we give the following definitions:

- TP: sum of pixels with values greater than the reconstruction threshold in the reconstructed solution map that were less than three pixels away from a source in the true solution map

- FP: sum of pixels with values greater than the reconstruction threshold in the reconstructed solution map that were equal to or greater than three pixels away from a source in the true solution map
- TN: sum of pixels with values lower than the reconstruction threshold in the reconstructed solution map that were also empty in the true solution map
- FN: sum of pixels with values lower than the reconstruction threshold in the reconstructed solution map that were not empty in the true solution map,

where TP refers to the true positives, TN refers to true negatives, FP refers to the false positives, and FN refers to false negatives.

Given that source-finding in the current work was defined as being directly related to the sum of pixels output by the source-finders, the sum of the sources detected between the source-finders was not expected to be constant.

Since the true catalog was quite richly populated with sources, and given the three pixel leniency, there could be some sources that were found across both algorithms by chance. Ideally, there should be zero chance findings, but in reality, there will be some small fraction. We used the SNR = 1 dataset to test this effect, as this dataset had the highest population of sources in the solution map (and also in the reconstructed map). Therefore, the SNR = 1 dataset could be considered to be the worst case scenario for chance matches. The effect of chance matches was tested by randomly rotating the reconstructed solution maps, comparing with the real solution map, and calculating the metrics, as for the rest of the results.

The precision, recall, and F1 score metrics, in the form of bar plots, are provided in the current work. We did not include the accuracy because of the way the true negatives were defined. The value was always very high, leading to accuracies greater than 99% across both source-finders. The metrics are defined in Equations (2)–(5).

$$\text{Precision} = \frac{\text{TP}}{\text{TP} + \text{FP}} \tag{2}$$

$$\text{Recall} = \frac{\text{TP}}{\text{TP} + \text{FN}} \tag{3}$$

$$\text{F1 score} = \frac{2 \times \text{Precision} \times \text{Recall}}{\text{Precision} + \text{Recall}} \tag{4}$$

$$\text{Accuracy} = \frac{\text{TP} + \text{TN}}{\text{TP} + \text{FP} + \text{TN} + \text{FN}}. \tag{5}$$

The original reconstructed image output of ConvoSource was composed of continuous pixel values that were mainly close to zero. To determine the output predictions for the source locations, we defined a reconstruction threshold that ranged between zero and one. Then, we chose the value across all metrics depending on which reconstruction threshold produced the highest F1 score. PyBDSF only produced a binary output depending on whether a source was found or not.

The bar plots of the F1 score, defined as the harmonic mean across precision and recall, are provided in the main text, as well as a subset of the confusion matrices. The precision and recall bar plots are given in Appendix A. In order to gauge the reliability of the predictions, we included the Kappa statistics [37], which measures the correlation of the predictions between the source-finders. Values of one indicate complete agreement and values of zero no agreement, or the agreement that would be expected by chance.

We also omitted the results across the B5 dataset because both source-finders failed to recover any sources across any integration time. This was most likely because although the noise was lower at higher frequencies, the sources tended to occupy fewer pixels. Furthermore, a higher frequency resulted in a lower surface brightness, and there were very few sources available in the catalog at the B5 frequency. For these reasons, source-finding at higher frequencies was more difficult.

### 3.1. Very Low Significance Source Metrics at SNR = 1

Figure 5 shows the F1 score metrics across the different classes of sources, as well as when all were considered together. ConvoSource almost always performed better across the SFGs and all sources in the B1 dataset, for all integration times, whereas PyBDSF performed better for the remaining datasets (SS and FS sources across B1 and B2 and SFGs and all sources at B2.)

The better performance of ConvoSource across the SFGs and all sources at B1 was most likely due to the effect of chance matches, as shown in Figure 6, which shows the source-finding metrics when the predicted source locations were randomized, to see how many sources were found due to chance. On average, ConvoSource was more affected by chance matches across the SFGs and all sources. Some possible causes of the increased chance matches in ConvoSource were that the SFGs were highest in number and that many sources found tended to be spread over several pixels rather than confined to one. On the other hand, PyBDSF was more affected on average by chance findings across the SS and FS sources. In ConvoSource, at worst, the chance matches reached up to ∼26% compared to real findings, whereas in PyBDSF, the effect was more pronounced in the datasets where fewer sources were found overall. The worst case for PyBDSF was across the SS sources in the B2 100 h dataset, where there were barely more real matches compared to chance matches. It should also be noted that the SNR = 1 dataset was the noisiest one that also had the most densely populated solution and reconstruction maps, which maximized the risk of chance findings, therefore representing the worst case scenario in terms of datasets. We further note that the sources had very low significance at this SNR.

An improvement in the F1 score across the SFGs could be observed due to augmenting the images containing all sources, since the vast majority of all sources were SFGs. However, augmenting the SS and FS sources did not improve the SFG scores by much, since we were not giving the network more examples of SFGs to train. Image augmentation did not have the same effect on the randomized data, as shown in Figure 6.

Figure 7 may indicate possible reasons why the ConvoSource performance was poorer overall compared to PyBDSF, at an SNR = 1. Since the solutions were injected into the map at the threshold of the mean background noise level, there appeared to be solutions that were not obvious by eye and could become confused with the background noise. It is therefore possible that ConvoSource did not successfully learn to extract the sources at this SNR. For the examples given, there was one SS source in each map, while the rest were SFGs. Both PyBDSF and ConvoSource recovered the SS source in the top row, whereas ConvoSource found a false positive and missed other SFGs. PyBDSF recovered one of the SFGs successfully, but missed the others. Neither PyBDSF nor ConvoSource recovered the SS source in the bottom row, and ConvoSource partially recovered one of the SFGs even though the location was spread out over several pixels. It missed the other SFGs and identified false positives. PyBDSF recovered only one SFG in this example and missed the others, which resulted in a number of false negatives.

Table 6 shows the correlations between the predictions given by the different runs of ConvoSource, compared with that given by PyBDSF. We note that in calculating the correlations, exact source locations between the source-finders was assumed (there was no leniency in the pixel locations). There were moderate correlations between the source-finders at an SNR = 1 across the B1 (560 MHz) dataset. The correlations tended to improve on average when ConvoSource was trained with the augmented datasets and when longer exposure times were used. In the B2 (1400 MHz) dataset, there were only random correlations at the shorter exposure times of 8 h and 100 h; however, the correlations were again moderate at 1000 h.

We note that at this SNR, the signal was on the same level as the noise, where more sources were available, and it represented the most difficult case in separating the pixels belonging to a source as opposed to noise. Furthermore, ConvoSource output the source locations spread over several pixels, whereas PyBDSF localized the sources to single pixels; therefore, even if the same source was found by

both source-finders, the correlations between them for one source in question may cancel out given the disagreement in the neighboring pixel values.

A likely reason for the high level of chance correlations in the B2 dataset was that although the noise was lower, the sources occupied fewer pixels. Furthermore, at higher frequencies, there were lower surface brightness values. It was for these reasons that there was more difficulty in finding sources at higher frequencies. We note that when the exposure time was increased from 100 h to 1000 h in the B2 dataset, this resulted in data that were greatly reduced in noise compared to increasing the exposure time from 8 h to 100 h. Therefore, the radio signals from the sources were in much higher contrast to the background noise. As a result, the correlations became moderate again.

**Table 6.** The Kappa statistics as measured using the correlation of the predictions between all runs of ConvoSource against PyBDSF, across all frequencies and exposure times at SNR = 1.

|  | **B1_8 h** | **B1_100 h** | **B1_1000 h** | **B2_8 h** | **B2_100 h** | **B2_1000 h** |
|---|---|---|---|---|---|---|
| ConvoSource_None | 0.30 | 0.56 | 0.55 | 0.0 | 0.0 | 0.32 |
| ConvoSource_Extended | 0.52 | 0.58 | 0.51 | 0.0 | 0.0 | 0.50 |
| ConvoSource_All | 0.33 | 0.61 | 0.53 | 0.0 | 0.0 | 0.41 |

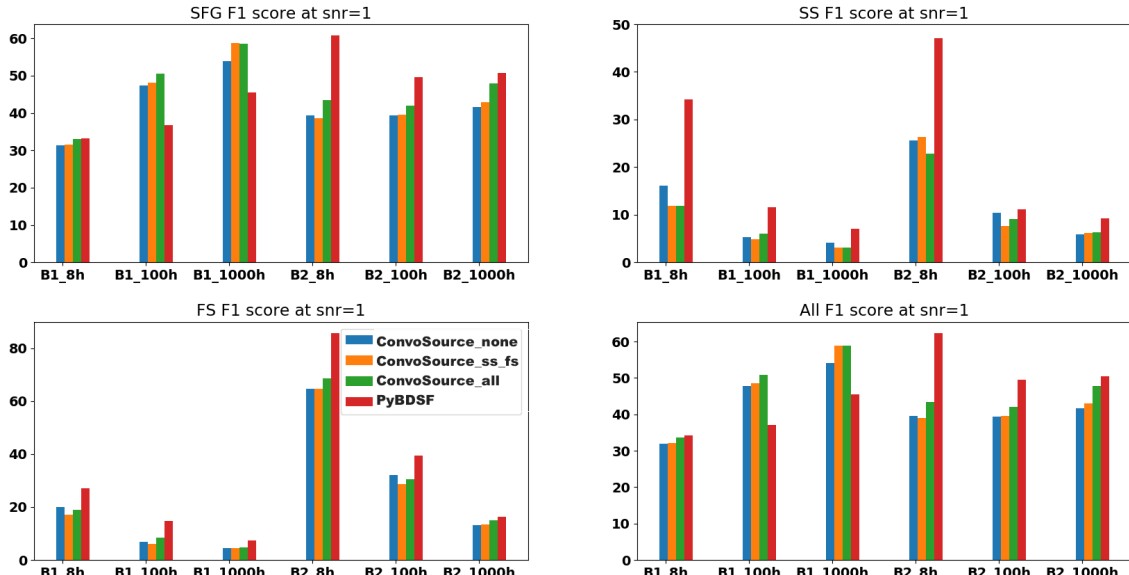

**Figure 5.** F1 scores at SNR = 1, across the two frequencies B1 (560 MHz) and B2 (1400 MHz) and the three integration times. There are three results given from ConvoSource, depending on the augmentation used when training. The blue bar represents no augmentation; orange represents augmenting the SS and FS sources; and the green bar represents augmenting all sources. The graphs show that PyBDSF usually performed better compared to ConvoSource at this SNR. Although it appeared that ConvoSource performed better across the SFGs and all sources in the B1 dataset, for all integration times, the better performance appeared to be explained by the increased proportion of chance matches at this SNR, as shown in Figure 6. However, it should be noted that these sources had very low significance given the SNR.

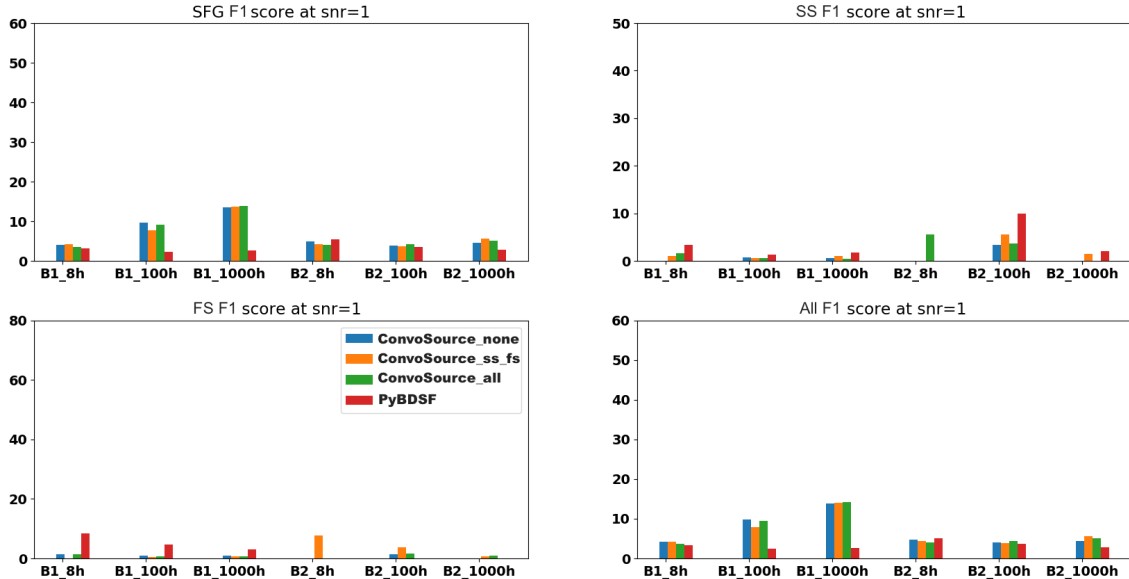

**Figure 6.** Showing the effect of randomly rotating the reconstructed matrix of source locations to investigate the proportion of chance findings.

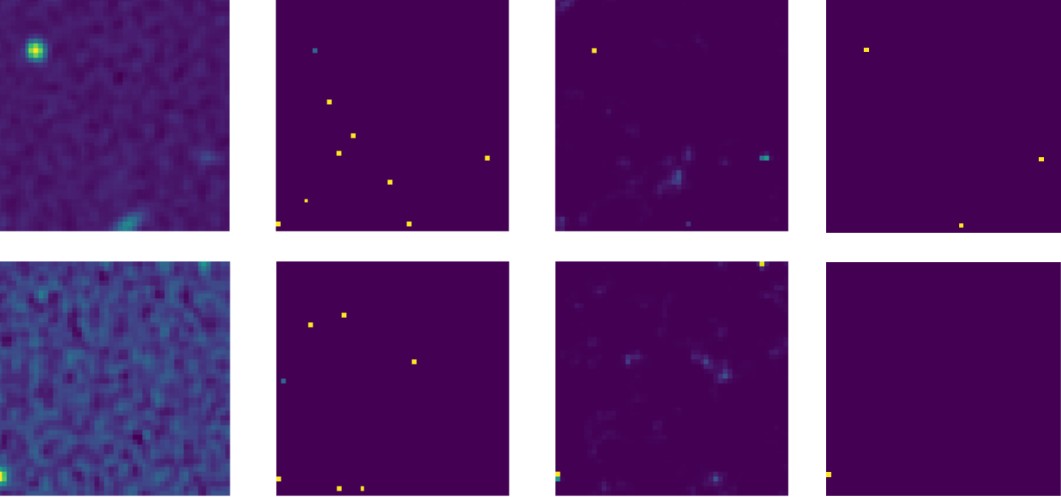

**Figure 7.** The top and bottom rows show a couple of examples of a real map for B1 at 8 h (first column), the solutions when injected into the map given the SNR = 1 threshold (second column), the predicted locations by ConvoSource after training on the original images only (third column), and the predicted locations by PyBDSF (fourth column).

### 3.2. Low Significance Source Metrics at SNR = 2

Figure 8 shows the F1 score metrics across the different classes of sources, as well as when we considered them all together.

Across the SFGs/all sources at SNR = 2, ConvoSource performed better on average, where now it recovered these sources better in the B2 8 h dataset, and one example is shown in Figure 9. However, PyBDSF generally performed better across the SS and FS sources.

Considering the F1 scores of the SS sources in the 8 h datasets, the augmentation of either the SS/FS or all sources worsened the score, most likely because this dataset was the noisiest of the three; therefore, some signal would become lost in the noise. There were generally slight improvements with the augmentation of the SS/FS at the other two integration times, as the noise was reduced. The SS sources were the ones that varied most in morphology (they had the greatest amount of extended

emission and gave rise to FRI/FRII type structures); however, there were not many original examples of these. Additionally, the signal threshold was set at only twice the noise threshold, so there were more solutions in the map, increasing the risk of sources being contaminated with noise. PyBDSF clearly outperformed ConvoSource across the SS sources.

Across the FS sources, there were two datasets in which ConvoSource performed better than PyBDSF (B2 at 8 h and B2 at 100 h); however, for the remainder, it did slightly worse than PyBDSF. The augmentation of the SS/FS sources always improved the F1 score across these FS sources; however, it did not always improve when augmenting "all" sources, since most of these sources were SFGs; therefore, proportionally, there were fewer SS/FS sources to train. We noted that when using ConvoSource, the performance was better across the FS sources as these sources had a more defined morphology, which tended to be more compact compared to that of the SS sources.

A similar pattern was observed at the SNR of two, as was observed at SNR = 1 in regard to the effects of augmentation, where augmenting sources of the same type resulted in improved metrics for those sources.

Table 7 shows the confusion matrices across the B1 and B2 datasets for the 8 h and 1000 h integration times, when comparing the test results after using ConvoSource trained on the augmentation of all sources, against PyBDSF. We excluded the true negative counts for brevity as this denoted the total number of pixels where there was no solution, as well as no predicted source. Given that ConvoSource sometimes produced reconstructed solutions that were spread over several pixels and that true positives were defined as matches that occurred over less than three pixels of the true solution locations, ConvoSource detected more true positives. However, it also detected more false positives compared to PyBDSF, but fewer false negatives. Therefore, it missed fewer sources compared to PyBDSF.

Table 8 shows the Kappa statistics between PyBDSF and the networks of ConvoSource, trained with different augmented datasets. Similar to what was observed at SNR = 1, the correlations were moderate in the B1 dataset and were due to chance in the B2 dataset at the shortest exposure times. On average, the correlations were improved compared to those observed at SNR = 1, most likely because the signal was twice as strong compared to the noise. We note however that this SNR was still relatively low, so the sources did not have much significance.

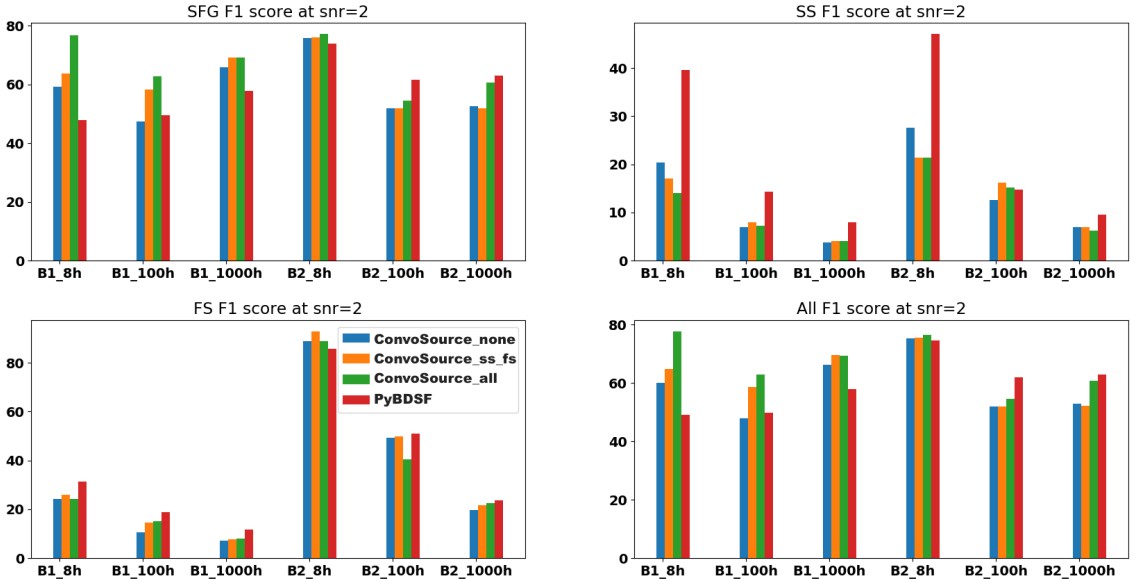

**Figure 8.** F1 scores at SNR = 2, across the two frequencies B1 (560 MHz) and B2 (1400 MHz) and the three integration times. There are three results given from ConvoSource, depending on the augmentation used when training. The blue bar represents no augmentation; orange represents augmenting the SS and FS sources; and the green bar represents augmenting all sources.

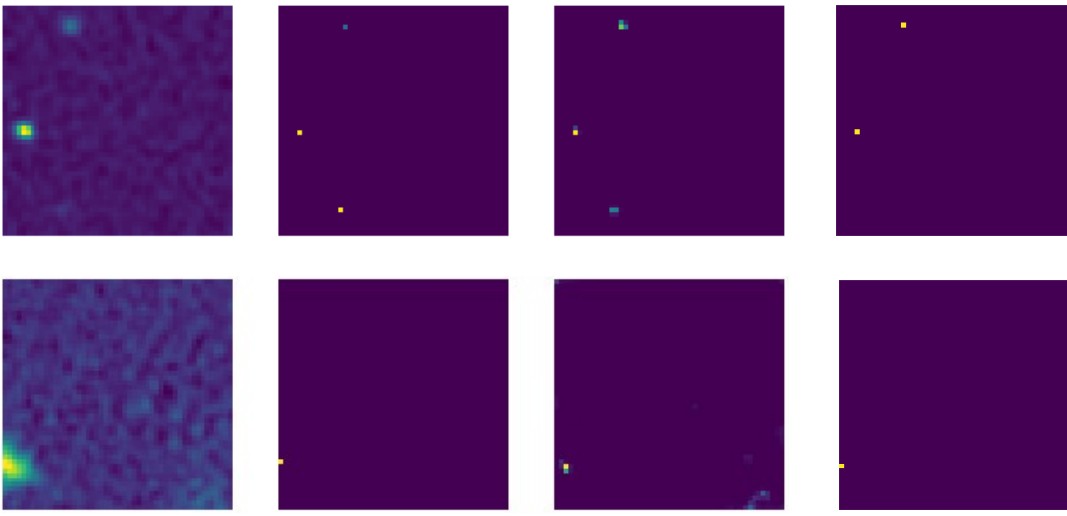

**Figure 9.** The top and bottom rows show a couple of examples of a real map for B2 at 8 h (first column), the solutions when injected into the map given the SNR = 2 threshold (second column), the predicted locations by ConvoSource after training on the original images only (third column), and the predicted locations by PyBDSF (fourth column).

**Table 7.** Showing all of the TP, FP, and FN across (3) ConvoSource augment all and (4) PyBDSF, at B1 8 h, B1 1000 h, B2 8 h and B2 1000 h, at an SNR = 2. The symbols in the first row (e.g., SFG_tp, _fp and _fn) represent the SFG true positives, false positives, and false negatives, respectively. The same pattern applies to the SS, FS, and all sources combined in the following 9 columns. The final two columns refer to the ratio of false positives to true positives and the ratio of false negatives to true positives.

| Method | SFG_tp | _fp | _fn | SS_tp | _fp | _fn | FS_tp | _fp | _fn | All_tp | _fp | _fn | #fp/#tp | #fn/#tp |
|---|---|---|---|---|---|---|---|---|---|---|---|---|---|---|
| B1_8 h | | | | | | | | | | | | | | |
| (3) | 1473 | 635 | 261 | 23 | 282 | 0 | 26 | 163 | 0 | 1522 | 561 | 316 | 0.37 | 0.21 |
| (4) | 314 | 73 | 611 | 19 | 57 | 1 | 8 | 35 | 0 | 341 | 46 | 663 | 0.14 | 1.94 |
| B1_1000 h | | | | | | | | | | | | | | |
| (3) | 5351 | 2735 | 2026 | 58 | 2722 | 0 | 68 | 1551 | 0 | 5477 | 2592 | 2235 | 0.47 | 0.41 |
| (4) | 3326 | 506 | 4333 | 57 | 1306 | 0 | 50 | 765 | 0 | 3433 | 429 | 4555 | 0.13 | 1.33 |
| B2_8 h | | | | | | | | | | | | | | |
| (3) | 628 | 52 | 319 | 3 | 22 | 0 | 12 | 3 | 0 | 643 | 56 | 340 | 0.09 | 0.53 |
| (4) | 130 | 13 | 79 | 4 | 9 | 0 | 9 | 3 | 0 | 143 | 8 | 89 | 0.06 | 0.62 |
| B2_1000 h | | | | | | | | | | | | | | |
| (3) | 2476 | 1593 | 1608 | 11 | 330 | 1 | 42 | 289 | 0 | 2529 | 1531 | 1734 | 0.61 | 0.69 |
| (4) | 1897 | 290 | 1932 | 12 | 226 | 1 | 31 | 199 | 0 | 1940 | 245 | 2050 | 0.13 | 1.06 |

**Table 8.** The Kappa statistics as measured using the correlation of the predictions between all runs of ConvoSource against PyBDSF, across all frequencies and exposure times at SNR = 2.

| | B1_8 h | B1_100 h | B1_1000 h | B2_8 h | B2_100 h | B2_1000 h |
|---|---|---|---|---|---|---|
| ConvoSource_None | 0.30 | 0.53 | 0.60 | 0.0 | 0.0 | 0.33 |
| ConvoSource_Extended | 0.30 | 0.62 | 0.57 | 0.0 | 0.0 | 0.29 |
| ConvoSource_All | 0.24 | 0.62 | 0.62 | 0.0 | 0.0 | 0.52 |

### 3.3. High Significance Source Metrics at SNR = 5

The opposite trend to what was observed at SNR = 2 was seen at SNR = 5, as shown in Figure 10, where now, PyBDSF performed better on the SFGs/all sources on average, whereas ConvoSource performed better on average on the SS and FS sources.

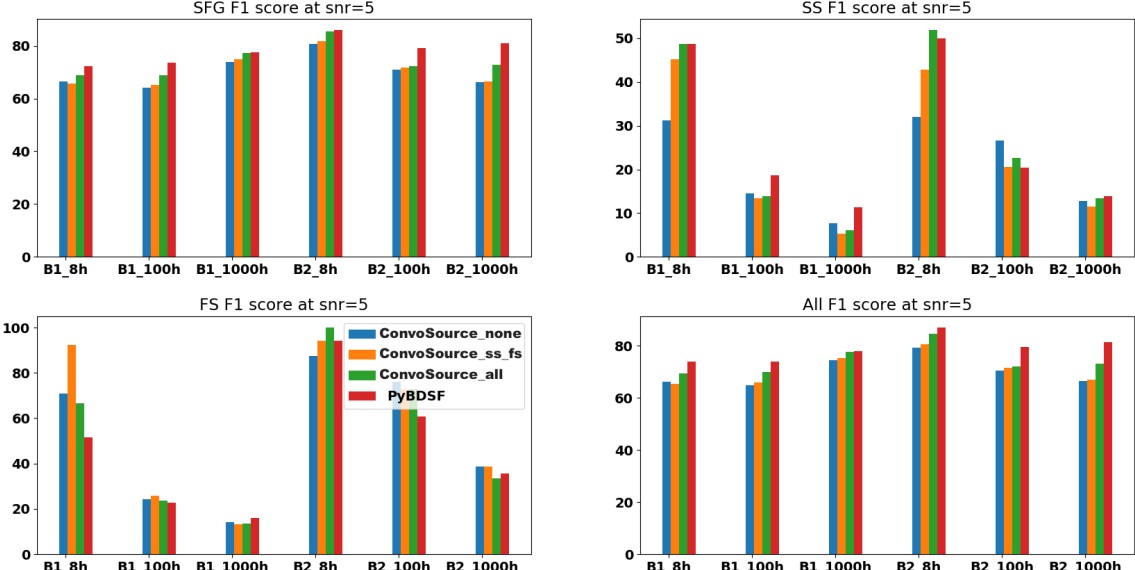

**Figure 10.** F1 scores at SNR = 5, across the two frequencies B1 (560 MHz) and B2 (1400 MHz) and the three integration times. There are three results given from ConvoSource, depending on the augmentation used when training. The blue bar represents no augmentation; orange represents augmenting the SS and FS sources; and the green bar represents augmenting all sources.

Therefore, when there was a higher signal to noise, ConvoSource could better extract the SS/FS sources compared to the SFG/point sources. For the majority of times, better results were achieved when augmenting either the SS and FS sources, or all; whereas when the signal to noise was lower, the performance of ConvoSource across these extended sources suffered, probably because the emission from them tended to become lost in the noise, whereas the SFGs were recovered better compared to when using PyBDSF at lower SNRs.

Figure 11 shows that ConvoSource can recover the SFG and SS source in the top row, as well as the SFG in the bottom row; however, at the expense of a couple of false positives. Meanwhile, PyBDSF did not recover any sources.

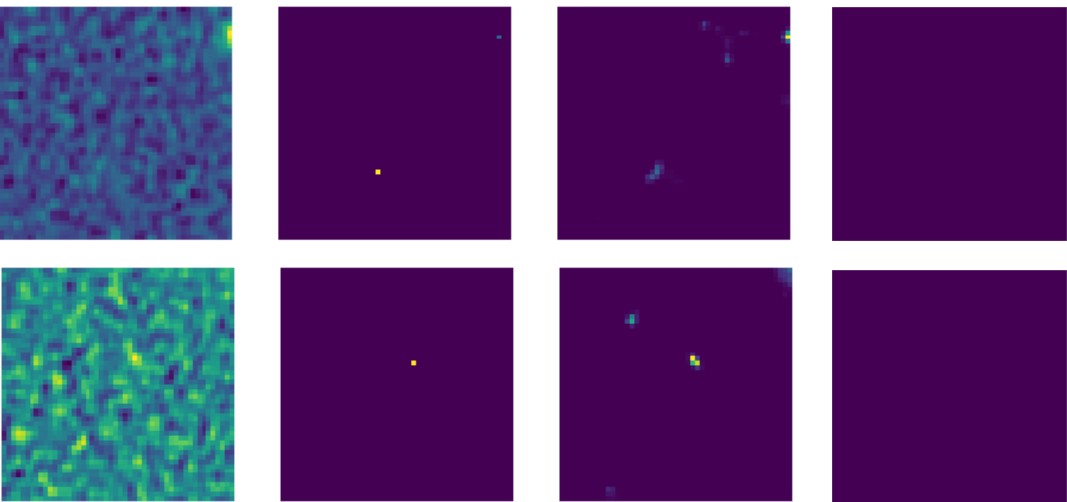

**Figure 11.** The top and bottom rows show a couple of examples of a real map for B2 at 8 h (first column), the solutions when injected into the map given the SNR = 5 threshold (second column), the predicted locations by ConvoSource after training on the original images only (third column), and the predicted locations by PyBDSF (fourth column).

Table 9 shows the confusion matrices at SNR = 5, for the same datasets and runs as was included for SNR = 2. There were fewer sources found by the source-finders overall as the SNR was higher compared to before; however, a similar trend was seen as before, where ConvoSource found more true positives and false positives, whereas PyBDSF found fewer true positives, but more false negatives. However, the ratio was not as pronounced when compared to what was observed at SNR = 2, as the signal-to-noise was now higher.

**Table 9.** Showing all of the TP, FP, and FN across (3) ConvoSource augment all and (4) PyBDSF, at B1 8 h, B1 1000 h, B2 8 h, and B2 1000 h, at an SNR = 5. The symbols in the first row (e.g., SFG_tp, _fp and _fn) represent the SFG true positives, false positives, and false negatives, respectively. The same pattern applies to the SS, FS, and all sources combined in the following 9 columns. The final two columns refer to the ratio of false positives to true positives and the ratio of false negatives to true positives.

| Method | SFG_tp | _fp | _fn | SS_tp | _fp | _fn | FS_tp | _fp | _fn | All_tp | _fp | _fn | #fp/#tp | #fn/#tp |
|---|---|---|---|---|---|---|---|---|---|---|---|---|---|---|
| B1_8 h | | | | | | | | | | | | | | |
| (3) | 444 | 175 | 225 | 20 | 41 | 1 | 11 | 11 | 0 | 475 | 164 | 256 | 0.35 | 0.54 |
| (4) | 304 | 60 | 172 | 18 | 38 | 0 | 8 | 15 | 0 | 330 | 40 | 193 | 0.12 | 0.58 |
| B1_1000 h | | | | | | | | | | | | | | |
| (3) | 3478 | 1422 | 629 | 35 | 1075 | 0 | 45 | 573 | 0 | 3558 | 1311 | 738 | 0.37 | 0.21 |
| (4) | 3070 | 663 | 1124 | 57 | 892 | 0 | 45 | 473 | 0 | 3172 | 554 | 1247 | 0.18 | 0.39 |
| B2_8 h | | | | | | | | | | | | | | |
| (3) | 332 | 44 | 70 | 7 | 13 | 0 | 8 | 0 | 0 | 347 | 47 | 80 | 0.14 | 0.23 |
| (4) | 128 | 13 | 29 | 4 | 8 | 0 | 8 | 1 | 0 | 140 | 8 | 34 | 0.06 | 0.24 |
| B2_1000 h | | | | | | | | | | | | | | |
| (3) | 1980 | 974 | 514 | 13 | 168 | 0 | 29 | 115 | 0 | 2022 | 923 | 567 | 0.46 | 0.28 |
| (4) | 1857 | 280 | 587 | 12 | 149 | 0 | 28 | 102 | 0 | 1897 | 224 | 637 | 0.12 | 0.34 |

Figure 12 shows the training and validation losses across the B1 and B2 frequencies, across all integration times. These losses were obtained using the binary cross-entropy cost function on the training and validation data, respectively, as a function of training epochs. In the left panel (B1 frequency; 560 MHz), the training and validation losses were roughly at the same level across the 8 h and 100 h integration times, whereas there was some underfitting observed in the 1000 h dataset. The underfitting generally indicates that a more complex architecture should be tried. In the right panel (B2 frequency; 1400 MHz), the 8 h integration time loss curves were at the same level, whereas there was some level of overfitting observed across the 100 h and 1000 h integration times. It was more difficult to find sources at the B2 frequency compared to B1 because the frequency was higher; therefore, the surface brightness of the sources was lower. It is interesting to note that for the same integration times across the two different frequencies, the same model tended to underfit on one dataset and overfit on the other. This indicated that using the same model across all frequencies and integration times was not ideal, that instead, each model should be tuned to the specific dataset at hand. Nonetheless, the resulting metrics were still competitive with those of PyBDSF and had the potential to outperform PyBDSF given a more optimally tuned model.

Table 10 shows the Kappa statistics between PyBDSF and the trained networks of ConvoSource, with different augmented datasets. The correlations were improved compared to those observed at an SNR = 2, between the B1 datasets at 100 h and 1000 h exposure time. However, they became random at the 8 h exposure time, which continued to be observed in the B2 dataset at 8 h and 100 h. Potential reasons for this behavior were that although the SNR was the strongest, there were fewer source location predictions given on average compared to at a lower SNR. Furthermore, there were cases for example in Figure 11, where PyBDSF failed to predict any location for the source; therefore, although ConvoSource may predict the correct location, it would not agree with the PyBDSF prediction.

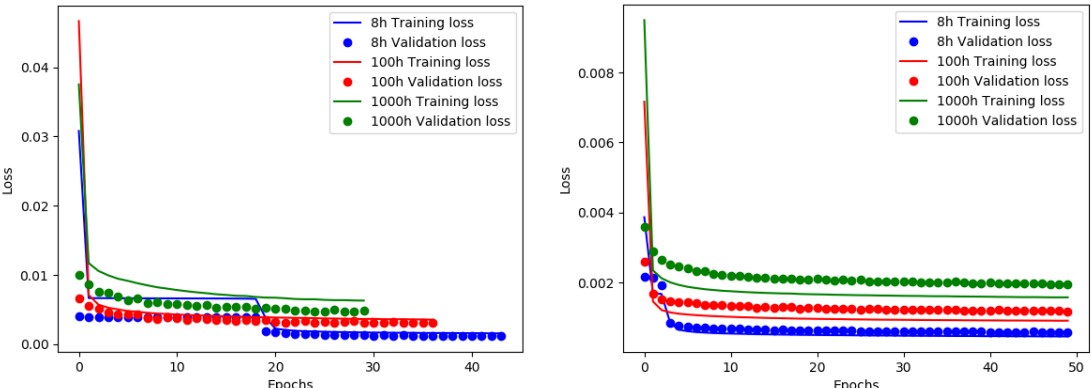

**Figure 12.** Training and validation losses across the three integration times at SNR = 5 across B1 and B2 datasets in the left and rights panels, respectively.

**Table 10.** The Kappa statistics as measured using the correlation of the predictions between all runs of ConvoSource against PyBDSF, across all frequencies and exposure times at SNR = 5.

|                      | B1_8 h | B1_100 h | B1_1000 h | B2_8 h | B2_100 h | B2_1000 h |
|----------------------|--------|----------|-----------|--------|----------|-----------|
| ConvoSource_None     | 0.0    | 0.52     | 0.67      | 0.0    | 0.0      | 0.1       |
| ConvoSource_Extended | 0.0    | 0.53     | 0.73      | 0.0    | 0.0      | 0.37      |
| ConvoSource_All      | 0.1    | 0.46     | 0.73      | 0.0    | 0.0      | 0.1       |

## 3.4. Execution Times

Both source-finders were run on a computing cluster using CPUs from 27 available Intel XEON CPU nodes, with a 3.5 GHz processor. There were six available cores per node.

Given that ConvoSource was built using Keras using the TensorFlow backend, it was possible to exploit the multicore architecture of the CPU and therefore speed up the generation and augmentation of images, as well as the training and testing time. Given the relatively small number of images and image sizes used in the current work, it was only necessary to use a single core at one time, on a node on the computing cluster. The use of multiple cores was possible by specifying the desired number in the configurations, using a command in Keras. The processes necessary for the computation would then be distributed evenly among the cores. On the computing cluster available, it should be possible, if necessary, to generate several hundred thousand images having a size of a couple of hundred pixels and perform training and testing on them, to use up the maximum computation of the six cores at a node in the cluster. For anything more intensive, it should be possible to configure the cluster to utilize multiple nodes.

In the current work, it was not necessary to exploit GPUs. They would be useful if larger sized blocks were desired and/or translating them by a smaller number of pixels (thus generating more original training images). GPUs would also be useful to speed up the generation of an increased number of augmented images and to try more complicated CNN architectures.

Table 11 shows the execution times required to generate segmented versions of the real maps and source location maps, augmented data, and training and testing times for ConvoSource. The times were compared to those obtained from running PyBDSF on the same area. The execution times across ConvoSource were subject to variability depending on how many sources there were to augment, as well as the total training time, which depended on the total number of images and epochs. The run where the SS and FS sources were augmented took a shorter time to train and test compared to the one where no augmentation was used because there were more epochs of training completed. The run that did not utilize augmentation was affected by the early stopping condition at an earlier point during training.

**Table 11.** Time required (in min) to generate segmented versions of the real and source location (solution) maps. For the different runs of ConvoSource, the time required to augment different sets of sources (SS and FS images, and all images) is given, as well as the corresponding times required for training and testing. We also show the amount of time needed to predict the source locations for PyBDSF. The number after the underscore in the first row of the table refers to the SNR at the given exposure time.

|  | 8 h_1 | 8 h_2 | 8 h_5 | 100 h_1 | 100 h_2 | 100 h_5 | 1000 h_1 | 1000 h_2 | 1000 h_5 |
|---|---|---|---|---|---|---|---|---|---|
| **B1** | | | | | | | | | |
| Generate solutions | 2.9 | 2.5 | 3.3 | 2.9 | 2.7 | 2.5 | 3.2 | 4.5 | 2.6 |
| Generate real data | 2.5 | 4.5 | 2.4 | 3.8 | 2.9 | 2.6 | 3.0 | 4.7 | 5.0 |
| Augment SS + FS | 0.2 | 0.2 | 0.1 | 0.6 | 0.4 | 0.1 | 1.8 | 1.1 | 0.3 |
| Augment all | 21.2 | 16.0 | 21.2 | 21.5 | 28.5 | 21.3 | 21.3 | 17.1 | 17.0 |
| Train none | 38.0 | 23.1 | 18.0 | 15.6 | 42.7 | 26.4 | 13.2 | 24.7 | 8.0 |
| Train SS + FS | 39.6 | 26.7 | 20.6 | 15.3 | 24.4 | 32.6 | 37.7 | 25.2 | 35.1 |
| Train all | 109.9 | 126.3 | 120.7 | 79.5 | 111.1 | 145.1 | 69.1 | 81.1 | 99.6 |
| Test none | 0.7 | 0.5 | 0.4 | 0.5 | 1.2 | 0.5 | 0.6 | 0.5 | 0.5 |
| Test SS + FS | 1.1 | 0.5 | 0.4 | 0.5 | 0.5 | 0.5 | 0.7 | 0.5 | 0.5 |
| Test all | 0.5 | 0.5 | 0.4 | 0.5 | 0.7 | 0.5 | 0.5 | 0.6 | 0.5 |
| PyBDSF | 5.2 | 5.7 | 5.7 | 6.2 | 6.3 | 5.2 | 19.1 | 18.4 | 20.1 |
| **B2** | | | | | | | | | |
| Generate solutions | 3.6 | 3.2 | 4.0 | 4.6 | 3.1 | 3.4 | 3.2 | 6.4 | 3.5 |
| Generate real data | 3.0 | 5.1 | 3.8 | 3.3 | 3.3 | 3.1 | 3.1 | 4.1 | 6.2 |
| Augment SS + FS | 0.1 | 0.1 | 0.0 | 0.2 | 0.2 | 0.0 | 0.2 | 0.2 | 0.1 |
| Augment all | 26.1 | 26.0 | 35.8 | 26.0 | 26.0 | 20.6 | 22.7 | 25.8 | 20.6 |
| Train none | 35.2 | 31.9 | 29.1 | 75.4 | 32.5 | 29.2 | 52.4 | 28.9 | 29.1 |
| Train SS + FS | 38.3 | 32.4 | 30.7 | 59.2 | 35.7 | 32.9 | 27.2 | 40.6 | 35.9 |
| Train all | 190.5 | 148.6 | 281.6 | 148.6 | 152.8 | 231.8 | 263.8 | 129.1 | 446.8 |
| Test none | 0.7 | 0.7 | 0.4 | 1.9 | 0.7 | 0.6 | 1.3 | 0.7 | 0.7 |
| Test SS + FS | 0.7 | 0.5 | 0.4 | 1.6 | 0.6 | 0.5 | 0.7 | 0.7 | 0.7 |
| Test all | 0.6 | 0.7 | 0.6 | 0.7 | 0.6 | 0.6 | 1.3 | 0.7 | 1.2 |
| PyBDSF | 8.6 | 6.8 | 5.9 | 23.2 | 21.5 | 22.2 | 21.2 | 22.1 | 21.4 |

## 4. Discussion and Conclusions

In the current work, we showed how the use of a simple CNN composed of three convolutional layers, a dropout layer, and a dense layer, as shown in Figure 1 and Table 1, could be competitive with a state-of-the-art source-finder, PyBDSF. Both approaches were tested across different frequencies, integration times, and signal-to-noise ratios, and the recovery metrics across the different source types of SFGs, SS-AGN, and FS-AGN sources were derived. The code used to obtain both the ConvoSource and PyBDSF results is available on GitHub[5]. Given that ConvoSource outputs continuous values in the reconstruction of the solution map, as defined by a reconstruction threshold that ranges between zero and one, whereas PyBDSF uses a fixed threshold, ConvoSource could be more flexible as a method as it attributes a probability to finding a source at a particular location.

ConvoSource also sometimes output the source location spread over a few pixels rather than being localized to a single one, which may provide additional information about the source. For example, it could be more extended or diffuse. The fact that ConvoSource spread out the source location over several pixels, which occurred more frequently at the lower SNRs and at shorter exposure times, where there were more sources present and their emission was more likely to get mixed with the noise, resulted in more true positives and fewer false negatives. However, at the same time, ConvoSource also produced a larger number of false positives compared to PyBDSF. A similar trend was seen at higher SNRs, although fewer true positives, false positives, and false negatives were found by both

---

[5] https://github.com/vlukic973/ConvoSource.

source-finders in comparison. For example, the SNR = 5 dataset had fewer solutions, but also the strongest signal. On the other hand, PyBDSF missed many more sources compared to ConvoSource, as the false negative counts were almost always higher.

It is interesting to note that the metrics across the SS and FS sources tended to be relatively low across both PyBDSF and ConvoSource. In fact, they decreased with increasing integration time, across all SNRs, with the dataset at the lowest frequency (B1) attaining the lowest metrics overall. Possible reasons could be that the SS and FS sources were smallest in number and their morphology was revealed as increasingly variable, as more extended emission was detected with the longer integration times.

In regard to how well the two methods extracted SFGs, SS, FS, and all source types combined across the SNRs, we saw that PyBDSF performed better on average compared to ConvoSource at SNR = 1. ConvoSource appeared to be more severely affected by chance matches at this SNR compared to PyBDSF; however, the sources had very low significance. In contrast, ConvoSource was better at extracting the SFGs and all sources at SNR = 2, whereas PyBDSF was better at extracting these at SNR = 5. ConvoSource was better at extracting the FS sources at an SNR of five, whereas PyBDSF was better for the FS sources at SNR= 2. ConvoSource was worse at extracting the SS sources at an SNR of 2; however, half the time, it was better than PyBDSF at extracting them at an SNR of 5.

We saw that image augmentation improved ConvoSource's performance when the relevant sources were augmented. Generating more "all" sources tended to improve the metrics across SFGs and "all" sources as these sources were largely made up of SFGs, and generating more SS and FS sources tended to improve their recovery, but not that of SFGs and all sources. Augmentation may also not work to improve the results as expected when the datasets are noisier, the sources are few in number, or if their morphology is ambiguous.

The reliability of the predictions was quantified using the Kappa statistics. At the lowest SNR of one, the correlations between the source-finder predictions tended to be moderate for most datasets; however, they were random in the B2 dataset at the shorter exposure times. This was likely due to the very low SNR since the signal was the same level as the noise, and it was the most populated dataset on average with source predictions. Furthermore, we note that ConvoSource tended to output the source locations over several pixels, whereas PyBDSF localized them to single pixels; therefore, a reduced Kappa statistic was likely given there would be a disagreement in values in the vicinity of a particular source. A general improvement of the Kappa statistics was observed with increasing SNR. Additionally, it should be possible to improve the Kappa statistics by simulating lower frequency data, increasing the exposure time, generating more augmented images, and by producing more images through using smaller spacings between the segmented blocks.

It should be noted that the Kappa statistic had several limitations. Although the statistic was designed to take into account the probability of chance agreements, it could not be directly interpreted, and the assumptions it made about rater independence were not well supported [37].

The run times indicated that PyBDSF took longer to run on average compared to when using a trained ConvoSource network. Despite the augmentation and training times of ConvoSource being longer on average, they only had to be done once for a particular dataset, after which the trained network could be used. We note that the running times could further be improved if multiple cores were used or it was run on a GPU.

Across the results for the low significance source metrics at SNR = 2 and high significance source metrics at SNR = 5, ConvoSource usually outperformed or had very similar performance metrics to PyBDSF across the shortest integration time datasets (8 h). This may indicate that it could more successfully model the noise at these SNRs and integration time compared to PyBDSF. The only times that ConvoSource performed visibly worse was in B2 at 8 h across the SS sources at an SNR = 2 and across all B2 at 8 h at SNR = 1. It appeared that ConvoSource had trouble modeling the noise as the SNR decreased, especially for sources with more extended emission. Potential ways to improve

the performance of ConvoSource at lower SNRs could be to use a more complex network and train for more epochs with a greater reconstruction threshold when using early stopping.

The injection of sources and, in turn, the ability to be found by the source-finders largely depend on the characterization of the background noise signal. In the current work, we used PyBDSF to estimate the background noise. Therefore, if there were more false negatives/positives, these missed/extra sources would be contaminating the background signal to some extent. Sources displaying a more compact morphology are unlikely to affect the background signal by much since the emission is localized to a very small area. However, the effect will be larger the more extended the source is. Some extended sources may have very faint and/or diffuse emission, which can mingle with the noise.

It appeared that ConvoSource performed better overall at larger SNRs and shorter integration times compared to PyBDSF, most likely because it had learned to model the noise in these images better and the sources showed a greater contrast against the background. The ratio of false positives to true positives was larger for ConvoSource; however, the ratio of false negatives to true positives was larger for PyBDSF. Therefore, ConvoSource and PyBDSF performed better in terms of recall and precision, respectively. As the SNR increased, ConvoSource became increasingly better at recovering the extended (SS and FS) sources and tended to outperform PyBDSF across most datasets at the highest SNR of five. However, at the same time, PyBDSF became increasingly better at recovering the SFGs and sources as a whole. With a decreasing SNR, ConvoSource was increasingly successful at recovering the more compact sources (SFGs) and all sources, whereas it performed worse with the extended sources, most probably because it had not successfully learned to extract the extended source signals from the noise at lower SNRs, on which PyBDSF did better.

Given that ConvoSource tended to perform better in terms of recall (as shown in Appendix A), overall compared to PyBDSF (therefore, it found fewer false negatives and hence picked up some sources that PyBDSF had missed), it could be used as part of a pipeline where ConvoSource is run first to find the sources, then PyBDSF is run to extract the precision values for these sources, perform further filtering, as well as characterize the sources.

The next step in developing ConvoSource would be to derive properties from the sources found. One way to do this may be to correlate the features detected by lower layers to the values given in the catalog, for example to match the total flux for a source in question to the emission detected by one of the feature maps. Previously, we attempted a regression technique to see if it could learn the continuous values provided in the catalog; however, our network failed to learn any property successfully. ConvoSource could also be made up of individual models that are targeted to the dataset at hand, where the training and validation losses are better matched. Another possible extension to the current work would be to train a CNN to learn to remove noise from data by generating 1000 h maps from 8 h, or 100 h ones.

**Author Contributions:** Conceptualization, V.L.; methodology, V.L.; software, V.L.; validation, V.L., F.d.G. and M.B.; formal analysis, V.L.; investigation, V.L.; resources, M.B.; data curation, V.L.; writing–original draft preparation, V.L.; writing–review and editing, V.L., F.d.G. and M.B.; visualization, V.L.; supervision, F.d.G. and M.B.; project administration, M.B.; funding acquisition, M.B. All authors have read and agreed to the published version of the manuscript.

**Funding:** We acknowledge funding by the Deutsche Forschungsgemeinschaft (DFG, German Research Foundation) under Germany's Excellence Strategy: EXC 2121 "Quantum Universe", 390833306.

**Acknowledgments:** We thank Anna Bonaldi, Hershal Pandya, Stijn Buitink, and Gregor Kasieczka for useful comments on the paper.

**Conflicts of Interest:** The authors declare no conflict of interest.

## Appendix A. Precision and Recall Graphs

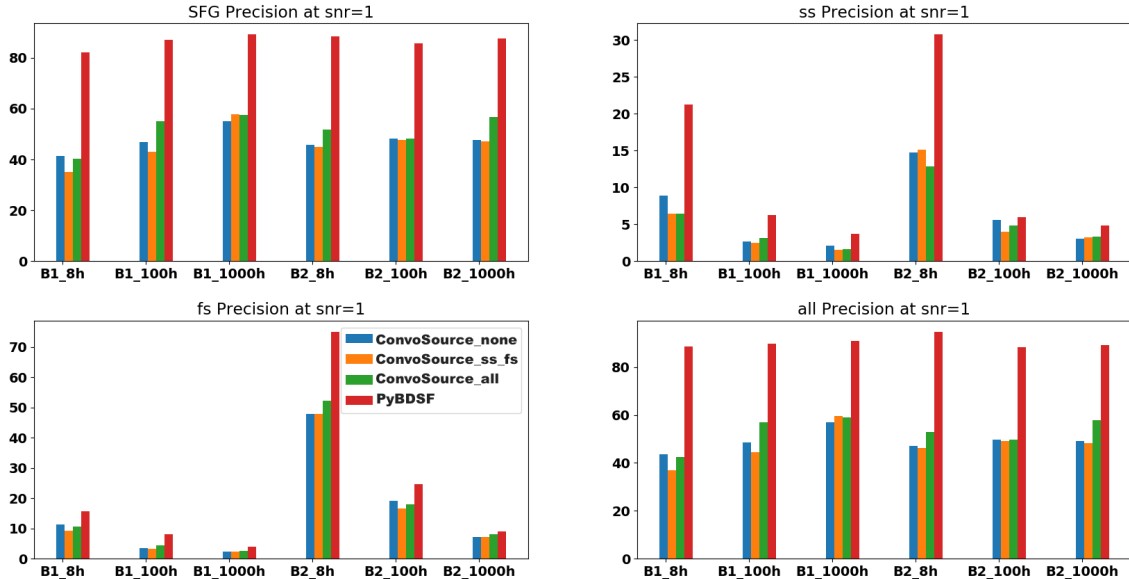

**Figure A1.** Precision values at SNR = 1.

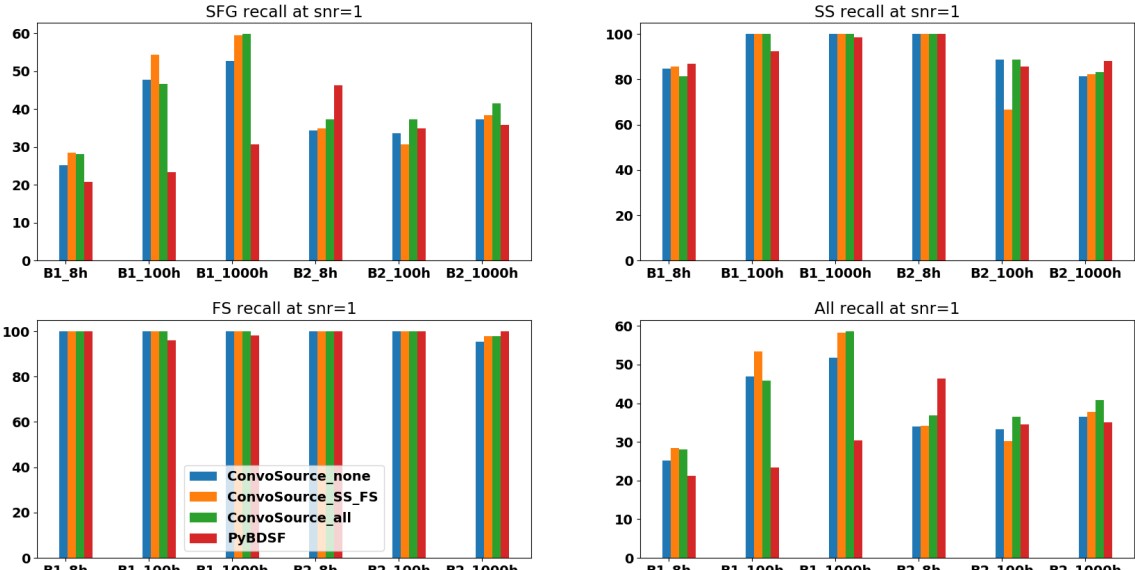

**Figure A2.** Recall values at SNR = 1.

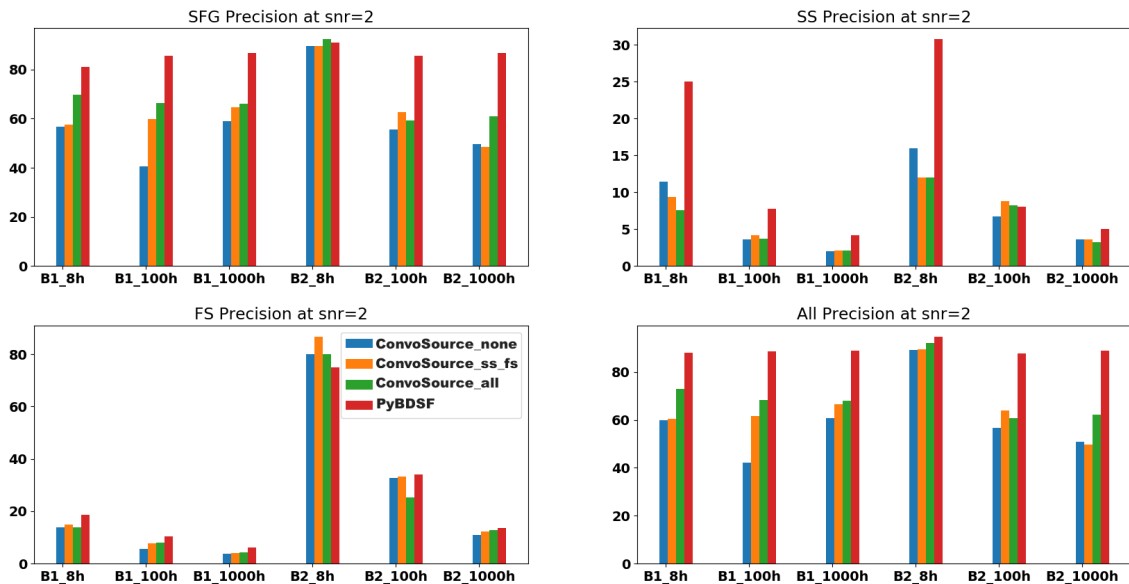

**Figure A3.** Precision values at SNR = 2.

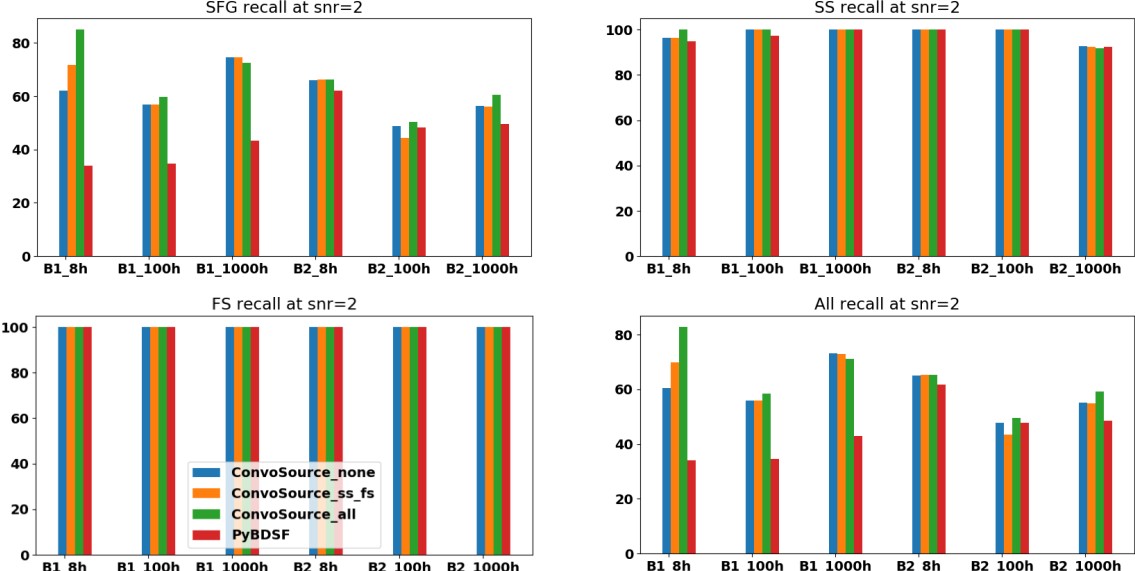

**Figure A4.** Recall values at SNR = 2.

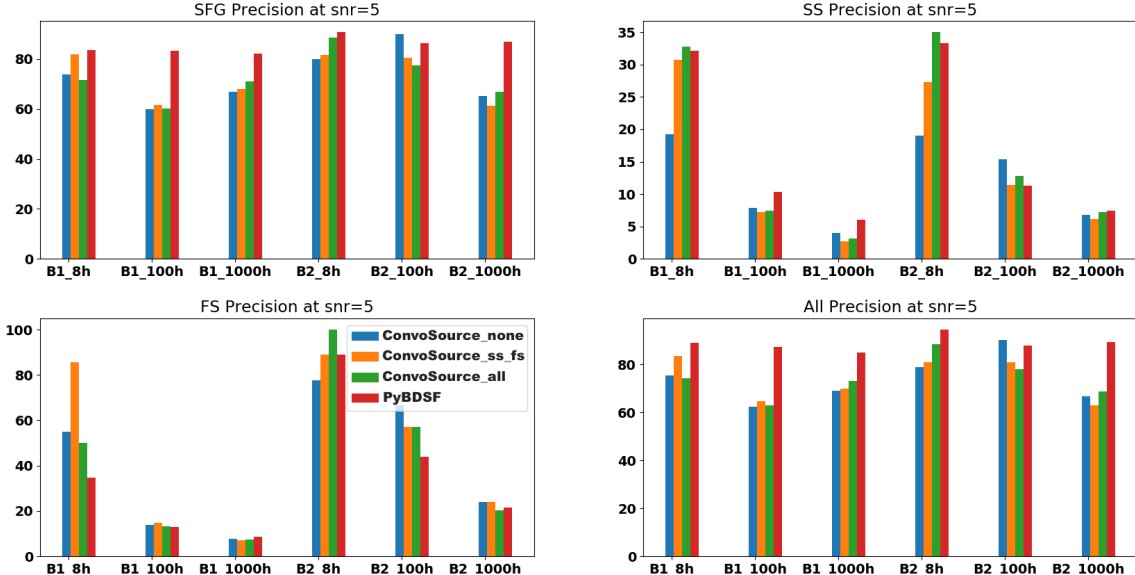

**Figure A5.** Precision scores at SNR = 5.

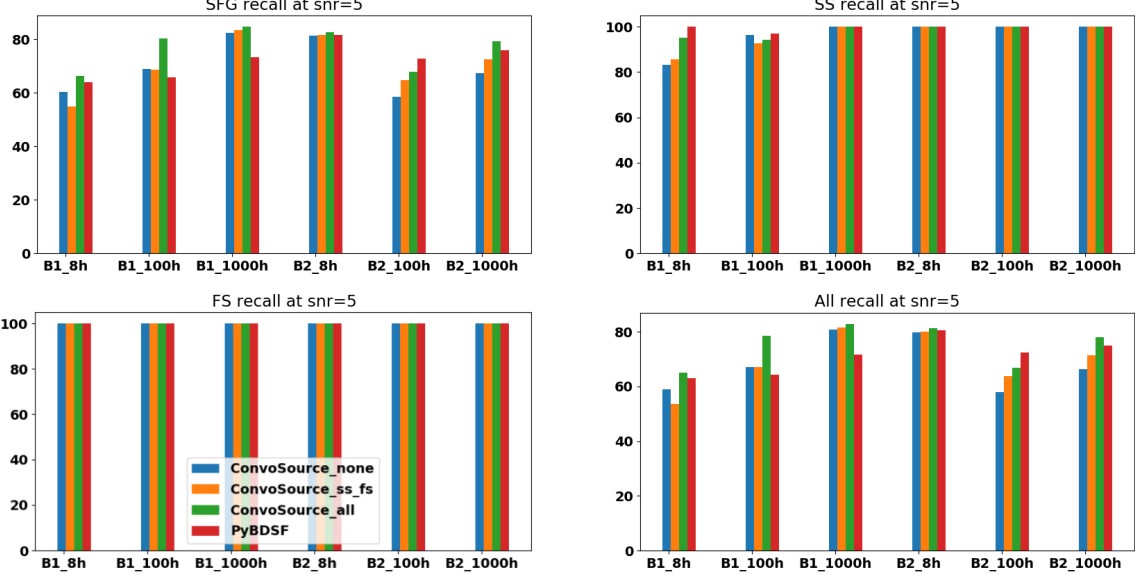

**Figure A6.** Recall scores at SNR = 5.

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
