# Peer review of "ConvoSource: Radio-Astronomical Source-Finding with Convolutional Neural Networks"

_galaxies, doi:10.3390/galaxies8010003_

Round 1

Reviewer 1 Report

The paper presents an application of Machine Learning to astrophysical radio images, targeting the SKA observatory. The authors adopt a Deep Convolutional Autoencoder to detect relevant objects, specifically star forming galaxies and AGNs, within noisy mock observations, mimicking expected SKA images. The performance of the resulting methodology is compared to that of a “traditional” approach. The outcomes are not conclusive, indicating that the effectiveness of the two approaches varies with the data and the kind of objects under investigation. However, the paper gives interesting evidences on how Machine Learning techniques can be useful for the next generation of radio astronomy data. It is properly written and it is proposed for publication with minor revisions, as detailed in what follows.   Detailed comments: Page 2, from line 69. A more general bibliography on machine learning applications in astrophysics, and specifically on source finding and classification, is necessary. In the last year the literature for the topic has grown exponentially and some review and credit are due. Line 78. The concept of hyperparameters has not been defined. Such concept is specific to Machine Learning and could be obscure to the non-expert reader. Please, explain what they are. Line 80. The “cost function and gradient-descent method” are not “parameters” of the deep learning method. Page 3, Section 1.2. It is not clear if in this section you want to describe in general all the possible types of radio sources or only those you target in the paper. An introductory, clarifying sentence would be beneficial. Section 1.3. The introduction to the deep learning and autoencoder methodology needs more details. Concepts like “convolutional layers”, “vanishing gradient problem”, fully connected neural networks”, “translational invariance” should be properly addressed and briefly explained for the non-expert reader. A figure explaining a prototype architecture of a deep learning and/or autoencoder network would be beneficial for the sake of comprehension. At line 101 I would not define as “stacking” the feature extraction process. Furthermore, I suggest to move the last sentence of the section (lines 114-115) to Section 2 (see also my next comment), and/or earlier in the introduction (with some more details), since it is an important statement that may be “lost” as it is now. Section 2.1. It would be useful to clarify here what is the main purpose of the network you developed (i.e. extract sources). From how it is written now, it is more clear what it is not (“subject of future work”) The rest of Section 2.1 would benefit of some reordering a prioritization. For instance, the last paragraph, describing an important feature of the network, should be placed earlier. Citing the GAN could be avoided (not really relevant for the context), or at least placed at the end of the Section. Line 247: dropout is used mainly to avoid "overfitting". I also suggest to clarify the purpose of introducing the fully connected layer. Figure 2 and 3. It is not clear what you are showing in the figures. Does the term “solutions” refers to the results of applying AutoSource to the data? If so, given that you are not classifying the different sources, how they can have different colors? This should be explained also at lines 282-283. Line 279: define what SS, FS and SFG stand for. From line 297 on. Please explain why you used only a subset of 4000x4000 pixels out of the full 32768x32768 pixels image. Justify also the choice of 50x50 pixels tiles. How was it set? Did you try also other sizes? What is the impact of the tile size on the results? Line 311. O(10e-6) in what units? Then, what kind of scaling are you applying to the data? Linear? Section 2.4. Why for augmenting the sources don’t you use the full ~32000x32000 map? Please explain. Line 345. “F1” is defined after it is used. Line 350: “that is less than 3 pixels of a source in true solution map”: there is something missing here. I think you mean that the distance is less than 3 pixels. Please rephrase. Paragraph starting at line 376. I suggest to place the description of pooling and its usage in Section 2.1. Seems more suitable there. “f1” in the titles of the figures 5, 6, 8 and 10 should be capital (“F1”) Figure 6: bars becomes really tiny. Figure 7 and 9. For the sake of clarity, increase the size of the panels as much as possible (maybe reducing also spaces between them). In the captions, I suggest to put (first column), (second column) etc. after the description of what they refer to. It is more clear. Tables 6 and 7. Define in the captions the symbols and describe better what they present. Line 479. Define what “validation losses” are. Section 3.4. I suggest to split the training time from the test time, since this can lead to a strong plus in favour of AutoSource. It is true that training time is quite long, but it must be done once, and then the network can be used for test data at a speed much higher than PyBDSF. Hence, in “working conditions” and more or less similar accuracy, the machine learning solution can be much more efficient than any other tool. This should also be highlighted in the conclusions (line ~543). Can AutoSource exploit the multicore architecture of the CPU? If yes, how? Can you comment on the scalability? Have you tried or at least, planned to try to exploit GPUs. Can you comment on that? Table 8: “across” is not the proper word there. Consider rephrasing. Line 554-556: I would recommend to remove the sentence, it weakens the conclusions.

Reviewer 2 Report

This paper trains a machine learning methodology in an attempt to locate sources in radio images.  The method is applied to simulated SKA data and compared to a more established algorithm.  In many of these simulated images, the recovery and accuracy, as measured via the F1 score, are largely similar, with each method doing better in certain circumstances.  However, the author's method is several times faster, an important advantage in the era of Big Data.  Overall, the English is good and results are presented clearly.

Major Comments:

The faster run time for AutoSource appears to be an important advantage given the amount of data expected from SKA.  Section 3.4 discusses this briefly, but between this and Table 8 only B1_8h run times are quantified.  I would recommend including run times for other bands / exposure times / SNR in the table as well.  As stated, "The execution times across AutoSource are subject to variability depending on how many sources there are to augment, as well as the total training time, which depends on the total number of images and epochs." Would it be possible to give a general quantification of these dependencies?

The quantification of effectiveness between the two methods is reliant upon the F1 score.  This is not a bad metric by any means, but it does leave some questions.  For example, in Figures 5, 6, 8, and 10, the F1 scores are shown to be different, but the significance of this difference is unclear as F1 does not yield an error.  To obtain an error, one could use a bootstrapping method by perturbing simulated images by the typical SNR and rerunning the algorithm.  However, I imagine this would be computationally expensive, and perhaps unreasonable.  Alternatively, one could also provide other metrics of information retrieval e.g., Matthews correlation coefficient, Cohen's kappa.  In particular, Cohen's kappa can quantify the effects of chance agreement, an issue mentioned several times in the work.

Minor comments:

SNR needs to be explicitly defined as signal-to-noise upon first usage in Abstract/Intro.

Lines e.g., 191, 268, 270, 328, 543, 588: Paragraphs should not be single sentences - put these sentences in other paragraphs.

Tables 2 & 3 can be easily combined for easier comparison.

Table 7 caption states that True Negatives are included, but I do not see them.

Table 8 - include other bands / exposure times / SNR as well.

Round 2

Reviewer 2 Report

The work has been made substantially clearer due to the additions from last submission. My only comment is that with the name change from AutoSource to ConvoSource, the authors should update Figures 5, 6, 8, 10, and those in the Appendix, which still are labeled AutoSource. Line 477 also uses AutoSource.

Author Response

Comment from reviewer:

The work has been made substantially clearer due to the additions from last submission. My only comment is that with the name change from AutoSource to ConvoSource, the authors should update Figures 5, 6, 8, 10, and those in the Appendix, which still are labeled AutoSource. Line 477 also uses AutoSource.

Response:

We have updated the name from AutoSource to ConvoSource in the figures mentioned, and in line 477.